## TECHNIQUES AND RESOURCES

# Integration of spatial and single-nucleus transcriptomics to map gene expression in the developing mouse kidney

Christopher P. Chaney[1,2,3,*], Alexandria N. Fusco[1,3], Elyse D. Grilli[1,3], Jane N. Warshaw[1,4], Peter M. Luo[3], Ondine Cleaver[3], Denise K. Marciano[1,4] and Thomas J. Carroll[1,3,*]

## ABSTRACT

The kidney is a complex organ requiring tightly coordinated interactions between epithelial, endothelial and mesenchymal cells during development. Congenital kidney defects can result in kidney disease and renal failure, highlighting the importance of understanding kidney formation mechanisms. Advances in RNA sequencing have revealed remarkable cellular heterogeneity, especially in the kidney stroma, although relationships between stromal, epithelial and endothelial cells remain unclear. This study presents a comprehensive gene expression atlas of embryonic and postnatal kidneys, integrating single-nucleus and *in situ* RNA sequencing data. We developed the Kidney Spatial Transcriptome Analysis Tool (KSTAT), enabling researchers to identify cell locations, predict cell–cell communication and map gene pathway activity. Using KSTAT, we were able to uncover significant heterogeneity among embryonic kidney pericytes, providing an important resource for hypothesis generation and advancing knowledge of kidney development and disease.

KEY WORDS: Single-nucleus RNA sequencing, Spatial transcriptomics, Kidney development, Cell communication

## INTRODUCTION

Recent advances in single-cell (scRNA-seq) and single-nucleus RNA sequencing (snRNA-seq) have revolutionized our understanding of cellular composition and signaling. These technologies have uncovered previously uncharacterized cell types, unexpected transcriptional states, and key insights into cellular differentiation. By simultaneously profiling all cells within a tissue, these approaches provide an unbiased view of cellular composition and behavior. This has led to a deeper understanding of relationships among cells, including transitions between cellular states during development, regeneration, and injury response. Rapid technological advancements now enable the analysis of tens of thousands of transcriptomes across various species, organs and experimental conditions.

[1]Department of Internal Medicine, Division of Nephrology, University of Texas Southwestern Medical Center, Dallas, TX 75390, USA. [2]Department of Internal Medicine, Division of Hospital Medicine University of Texas Southwestern Medical Center, Dallas, TX 75390, USA. [3]Department of Molecular Biology, University of Texas Southwestern Medical Center, Dallas, TX 75390, USA. [4]Department of Cell Biology, University of Texas Southwestern Medical Center, Dallas, TX 75390, USA.

*Authors for correspondence (christopher.chaney@utsouthwestern.edu; thomas.carroll@utsouthwestern.edu)

C.P.C., 0000-0002-4144-4180; T.J.C., 0000-0002-8322-4928

However, an important aspect of cellular heterogeneity remains poorly understood: the spatial location of transcriptionally diverse cell states within their native tissue environment. scRNA-seq relies on tissue dissociation for microfluidic capture, which results in the loss of spatial information. This limitation hinders our ability to reconstruct the complex cellular landscape and its dynamics accurately.

The spatial location of a cell determines how it interacts with neighboring cells, responds to biochemical cues, and accesses essential nutrients and oxygen from the bloodstream. Integrating spatial information with transcriptome data can yield insights into cellular behavior, interactions and tissue organization. Spatial transcriptomics offers the potential to provide high-resolution information about tissue composition and function.

The kidney is a highly complex organ that plays a crucial role in maintaining blood chemistry. It achieves this through functional units known as nephrons, epithelial tubules patterned along a proximal-distal axis into distinct functional domains. An adult human kidney contains, on average, one million nephrons per kidney. Defects in the patterning and differentiation of the kidney and ureter can impair renal function, potentially necessitating dialysis or transplantation to sustain life. Congenital anomalies of the kidney and urinary tract (CAKUT) account for approximately 10% of all identified birth defects (Tain et al., 2016). Studies across populations estimate the prevalence of CAKUT ranges from 1.6 to 20 cases per 1000 births (Li et al., 2019; Hays et al., 2022). Additionally, deficits in nephron number have been implicated in systemic hypertension (Brenner and Mackenzie, 1997) and chronic kidney disease (Good et al., 2023). A deeper understanding of kidney development is therefore essential for improving the diagnosis and treatment of birth defects and for addressing subtler contributions to systemic disease arising from aberrant kidney development.

The developing kidney is a complex organ composed of multiple cell types derived from at least four distinct developmental lineages, with intricate interactions between epithelial, endothelial and mesenchymal cells (Carroll et al., 2005; Karner et al., 2011; Schnell et al., 2022). Recent scRNA-seq studies have highlighted molecular heterogeneity among embryonic renal interstitial cells (England et al., 2020), suggesting that these diverse populations create unique microenvironments that influence the differentiation of adjacent epithelial and endothelial cells.

The cellular complexity of the kidney poses significant challenges for researchers investigating its form and function. Analyses that lack spatial context risk misinterpretation of data, particularly in studies of cell–cell signaling. Recent efforts have aimed to create spatial atlases of gene expression in the embryonic kidney (Little et al., 2007; McMahon et al., 2008; Harding et al., 2011; Yu et al., 2012; Lindström et al., 2021; Fusco et al., 2025). While valuable, these resources are often constrained by the resolution of molecular localization technologies and the breadth or depth of gene coverage,

meaning the number of genes detected or the number of transcripts detected per gene.

In this study, we address these limitations by combining single-cell transcriptomics with high-resolution spatial transcriptomics to generate a virtual atlas of embryonic and early postnatal mouse kidney tissue. Our analysis emphasizes the recently characterized heterogeneous renal stroma (England et al., 2020; Fusco et al., 2025; Combes et al., 2019), accurately predicting the expression and activity of genes and pathways beyond the initial set of landmark genes. This approach enables spatial visualization of gene expression, gene set enrichment, transcriptional program activity, and cell-type annotation on tissue sections. Our framework integrates spatial information into downstream modeling and computational analysis, applicable to any tissue or dataset with scRNA-seq or snRNA-seq data and limited spatial transcriptomic data. The resolution and depth provided in this analysis allow researchers to uncover cellular and molecular processes influenced by the tissue microenvironment.

## RESULTS

### In situ mRNA sequencing on embryonic kidneys

To achieve a systems-level understanding of how the stroma and parenchyma interact during kidney development, we sought to characterize the relationships among all cell types in the kidney at discrete developmental stages while considering the organ's complex spatial architecture.

To accomplish this, we analyzed scRNA-seq datasets from embryonic day (E) 18.5 mouse kidneys (England et al., 2020; Combes et al., 2019). Using the EIGEN algorithm (Chaney et al., 2022), we identified high-quality marker genes representing distinct cellular populations or states within the dataset (Fig. S1). We selected 90 landmark genes, including multiple markers for each cell cluster, to ensure comprehensive coverage of all known cell types in the E18.5 mouse kidney. These included both previously described and novel markers. Notably, the selected genes were enriched for stromal cell markers, reflecting our specific research focus. A heatmap of the chosen landmark genes (Fig. 1) illustrates distinct expression patterns across cell types, providing a robust foundation for further investigation. To provide some relative insight, we have grouped the clusters into broad categories: endothelium, epithelium, interstitium, leukocyte and podocyte. Each classification includes multiple clusters that represent not only different cell types but also different developmental states. For instance, the nephron progenitor cells, although not technically epithelial, are included in the epithelial classification, and the podocyte cluster most likely includes parietal epithelial cells as well as podocytes.

We next performed direct RNA in situ sequencing (ISS) (Lee et al., 2022) on multiple sections of mouse kidneys at E15.5, E18.5 and postnatal day (P) 3 using fluorescently tagged barcodes for the 90 landmark genes. This method enables direct RNA detection in tissues by combining padlock probes and rolling circle amplification with ISS chemistry. Of the 90 targeted genes, 88 produced measurable signals across all three time points. Representative data from each time point are shown in Fig. S2A-C.

We analyzed scRNA-seq and snRNA-seq datasets from mouse kidneys at corresponding developmental stages. Previously published scRNA-seq data for E15.5 mouse kidneys (Lawlor et al., 2019; Naganuma et al., 2021) were obtained from Gene Expression Omnibus (GSE118486 and GSE149134, respectively). For E18.5 and P3, we generated our own datasets, as detailed in the Materials and Methods section. Following standard preprocessing, we obtained 5339 transcriptomes from E15.5 kidneys, 15,389 from

E18.5 and 10,008 from P3. Of the original 90 landmark genes, 85, 86 and 80 genes, respectively, had sufficient spatial and single-nucleus transcriptomic data to support further analysis.

### Probabilistic, spatial mapping of snRNA-seq data onto embryonic kidney sections: KSTAT

Using a statistical model that predicts the likelihood of a cell's location based on its landmark gene expression profile, we integrated transcriptome measurements from individual nuclei with their physical locations on reference kidney sections. This method was an adaptation of work previously performed by Satija et al. (2015) (see Materials and Methods).

This approach enabled us to map transcriptional data from every cell or nucleus sequenced onto a spatial framework (in this case, a kidney section), effectively generating a virtual in situ hybridization image for every gene expressed in the datasets. Because transcriptional data from every cell in a dataset can be probabilistically mapped to a spatial location, the information contained within the individual cell's transcriptome can also be probabilistically mapped. We refer to this resource as the Kidney Spatial Transcriptome Analysis Tool (KSTAT).

To assess the accuracy of our mapping approach, we compared actual gene expression measurements from ISS to the predictions generated by our model for the landmark genes. Since the landmark genes were selected based on clusters identified in E18.5 transcriptomic data, we focused our evaluation on this time point. The model assigns non-zero expression probabilities to multiple locations within the kidney section, resulting in diffuse calculated signals. To facilitate visual comparisons, we applied a threshold to the predicted expression data, selecting the highest-probability locations for display. The number of selected locations matched those with measured ISS signals. We observed a strong correlation between the measured and predicted gene expression data; however, some discrepancies were noted. For example, although Podxl mRNA is expressed in both podocytes and endothelial cells at E18.5, our predictions captured podocyte expression but under-represented endothelial expression. We hypothesized that these differences arose from setting the expression threshold too high. To address this, we developed two solutions: (1) an automated method for optimizing thresholds (described in the Materials and Methods section) and (2) a user-controlled threshold adjustment feature, analogous to varying the development time in colorimetric mRNA in situ experiments. Fig. 2 shows predicted expression data and the effects of applying three arbitrarily chosen thresholds (90%, 95% and 99%) for four representative genes. Overall, thresholds in the 95-99% range most accurately reproduced the measured signals.

We quantitatively evaluated the performance of our model by assessing the reconstruction error for each landmark gene using both the full model and 'leave-one-out cross-validation' (LOOCV), a standard machine-learning approach for estimating model generalizability to unseen data (Murphy, 2012). The first analysis assessed the model's ability to reconstruct measured gene expression using all available data, while LOOCV provided an estimate of the model's predictive capacity for genes excluded from the training set. To quantify uncertainty and assess the fidelity of each measurement, we estimated the probability of obtaining the observed expression or a more extreme value if the signal were randomly distributed throughout the reference space (Table S1). This was achieved by calculating the distance between the measured and inferred expression distributions and comparing these values to the distances measured for a bootstrapped random distribution. Using a lower tail probability threshold of 5%, we accurately

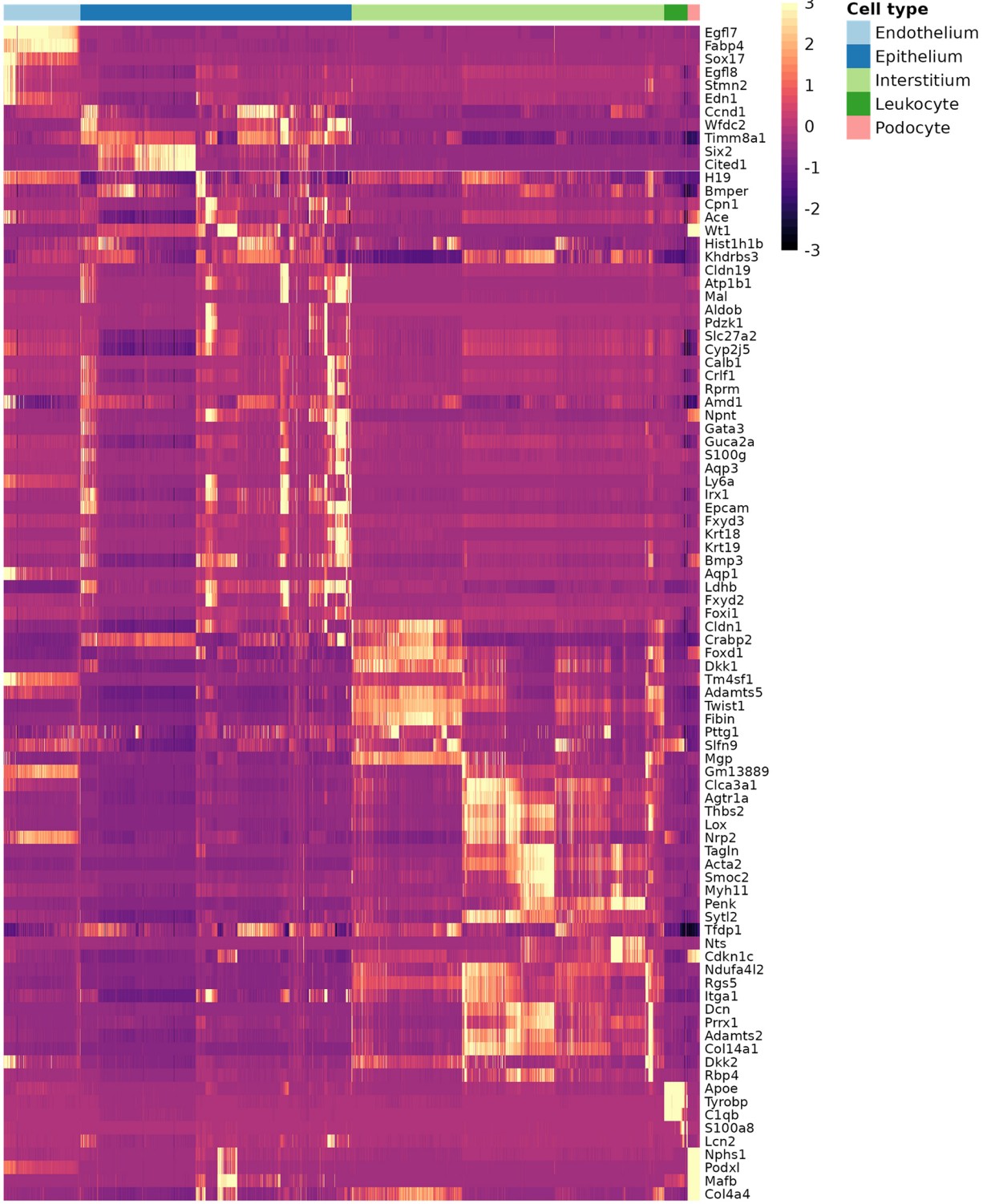

**Fig. 1. Expression of landmark genes across cell types.** A heatmap showing the relative expression levels of landmark genes across different cell types. The data are from a combined single-cell sequencing dataset from E18.5 kidney (GEO accession numbers GSE108291 and GSE155794). The expression values are imputed and normalized to account for technical variability in the data. Cell types are annotated and ordered along the *x*-axis, as shown by the color key at the top.

reconstructed spatial gene expression for 86 of the 88 landmark genes. LOOCV revealed reconstruction errors for four genes, resulting in accurate spatial predictions for 84 of 88 genes, an estimated accuracy of 95% when predicting spatial distributions for genes without spatial transcriptomics data.

We further validated our approach by comparing the imputed expression of several non-landmark genes with results from standard colorimetric *in situ* hybridization using single-probe sections. Predicted expression patterns qualitatively matched those obtained using antisense *in situ* hybridization techniques (Fig. 3A). Fig. S3

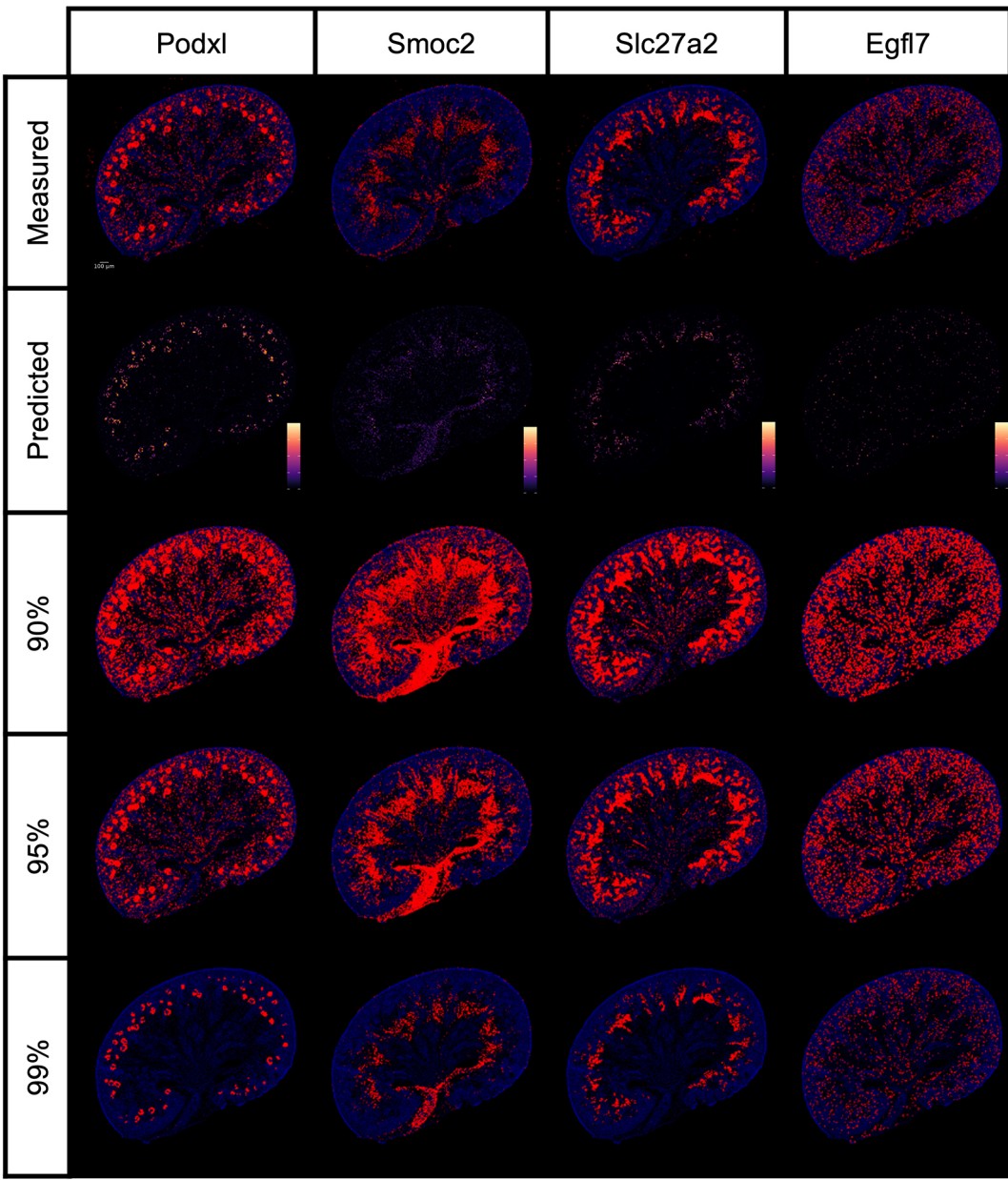

**Fig. 2. Thresholding reconstruction and thresholding.** Comparison of the transcript distributions measured by Cartana with those inferred by KSTAT for four genes: *Podxl*, *Smoc2*, *Slc27a2* and *Egfl7*. For each gene, the expected expression calculated using KSTAT is shown alongside the results of applying thresholds at the 90th, 95th and 99th percentiles.

shows predicted expression data for these genes at both E15.5 and P3, demonstrating consistency across developmental stages.

### KSTAT can resolve spatial differences in individual transcripts as well as cell clusters, revealing previously unappreciated heterogeneity in renal mural cells

The raw ISS data provides subcellular resolution, with a pixel edge length of 0.16 μm, surpassing the resolution of traditional *in situ* hybridization methods. For visualization, the diameter of each plotted point was adjusted to enhance aesthetic clarity. Fig. 3B-D shows uniform manifold approximation and projection (UMAP) plots for three genes – *Gucy1a2*, *Clca3a1* and *Cldn11* – within distinct clusters of interstitial cells. Previous investigations in our lab, constrained by the resolution limits of standard colorimetric *in situ* hybridization, were unable to determine whether these

genes exhibited overlapping expression patterns. Fig. 3E shows a simultaneous visualization of the expression of all three genes, with point diameters approximating the average mammalian cell size (15 μm) to ensure that non-overlapping points represent distinct cells. Using KSTAT, we demonstrate that *Gucy1a2*, *Clca3a1* and *Cldn11* exhibit non-overlapping expression patterns (Fig. 3F-H), underscoring the utility of our approach for resolving spatial relationships at high resolution.

In addition to mapping individual transcripts, our resource allows for the spatial mapping of transcriptional clusters. Recent studies using traditional methods have revealed transcriptional heterogeneity in endothelial cells of the embryonic kidney (Barry et al., 2019; Daniel et al., 2018). However, similar analyses have not been conducted for vascular-associated cells, such as mural cells. Fig. 4A shows a UMAP of isolated stromal cells,

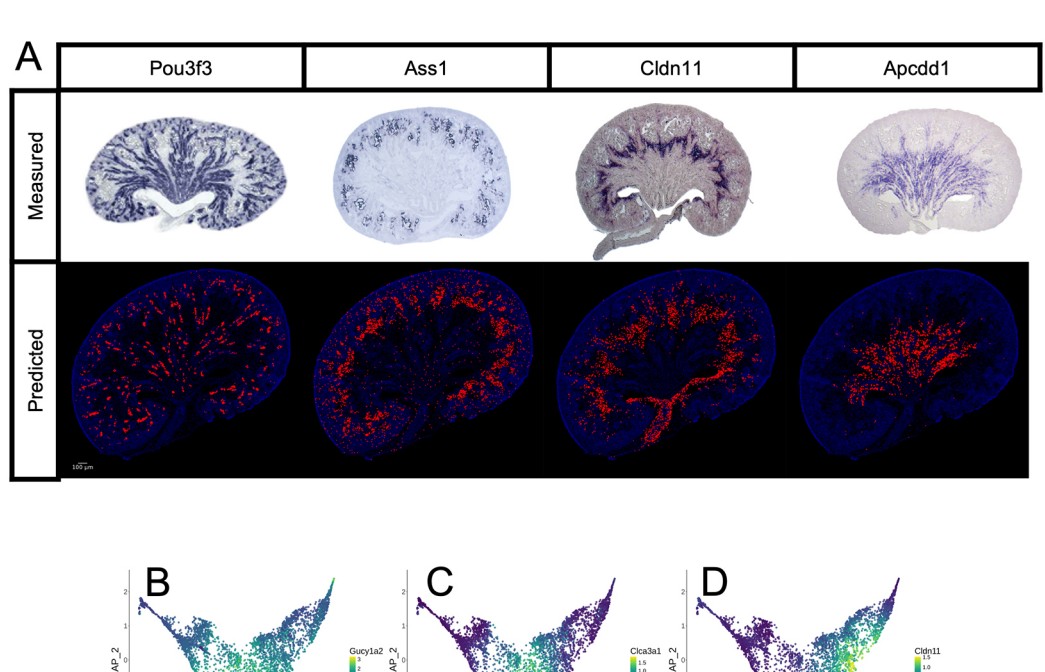

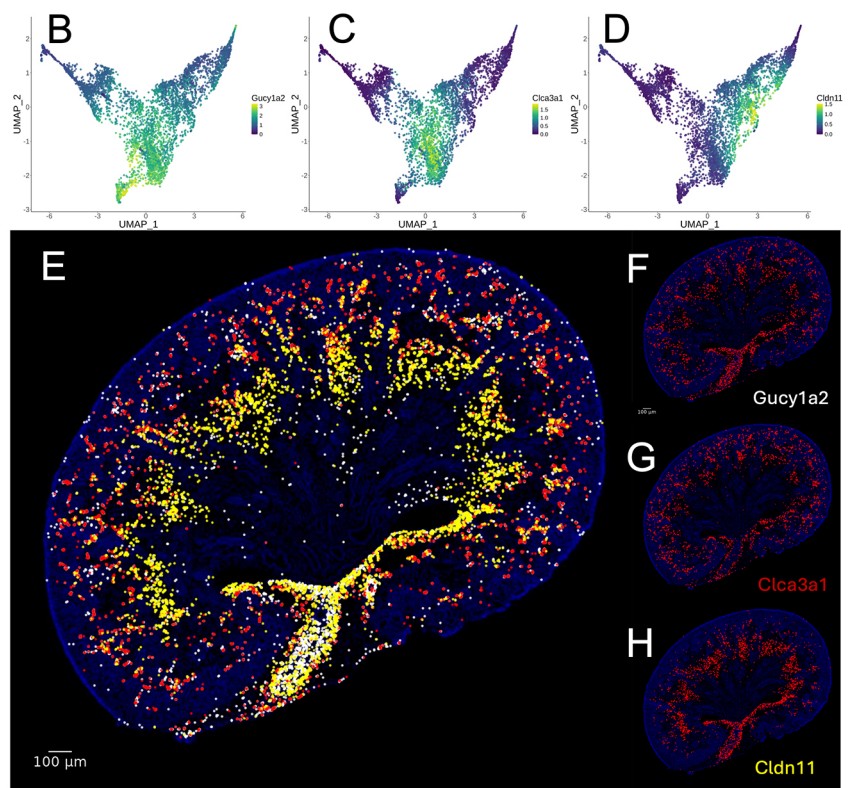

**Fig. 3. Multiplexed prediction of high-resolution spatial expression for unmeasured genes.** (A) Validation of our model's ability to predict gene expression patterns for unseen genes. The predicted spatial distributions of *Pou3f3*, *Ass1*, *Cldn11* and *Apcdd1* are compared to their corresponding expression patterns measured using colorimetric *in situ* hybridization in E18.5 kidneys. Notably, these four genes were not part of the training set, demonstrating our model's capacity to generalize to new genes. (B-D) UMAPs of isolated interstitial cells depicting expression of *Gucy1a2*, *Clca3a1* and *Cldn11*, which are differentially enriched in three separate clusters of these cells. (E) Simultaneous projection of predicted expression for *Gucy1a2*, *Clca3a1* and *Cldn11* onto the 2D kidney permits appreciation of spatial relationships between multiple genes with related yet distinct expression patterns (*Gucy1a2*, white; *Clca3a1*, red; *Cldn11*, yellow). (F-H) Individual projections of predicted expression for *Gucy1a2*, *Clca3a1* and *Cldn11*.

highlighting *Pdgfrb* expression, a canonical mural cell marker. *Pdgfrb* transcripts are enriched in four distinct transcriptional clusters. Fig. 4B-E displays UMAPs individually highlighting each of these clusters. The spatial relationships of these different clusters are unknown. To address this, we used KSTAT to map the clusters simultaneously (Fig. 4F) and individually (Fig. 4G-J) onto our E18.5 reference section. These data revealed significant spatial heterogeneity among mural cells. To validate this finding, we identified transcripts differentially enriched across the four

clusters (*Mgp*, *Cspg4*, *Acta2* and *Abcc9*). UMAPs representing expression of these genes are shown in Fig. S4A-D. Spatial mapping with KSTAT, in combination with the expected expression of canonical mural cell markers (*Rgs5* and *Pdgfrb*), with KSTAT confirmed the spatial differences in gene expression (Fig. S4E-K).

To corroborate these findings, we performed multiplex fluorescent *in situ* hybridization on E18.5 kidney sections using probes for the identified clusters (Fig. S4L-P). This analysis

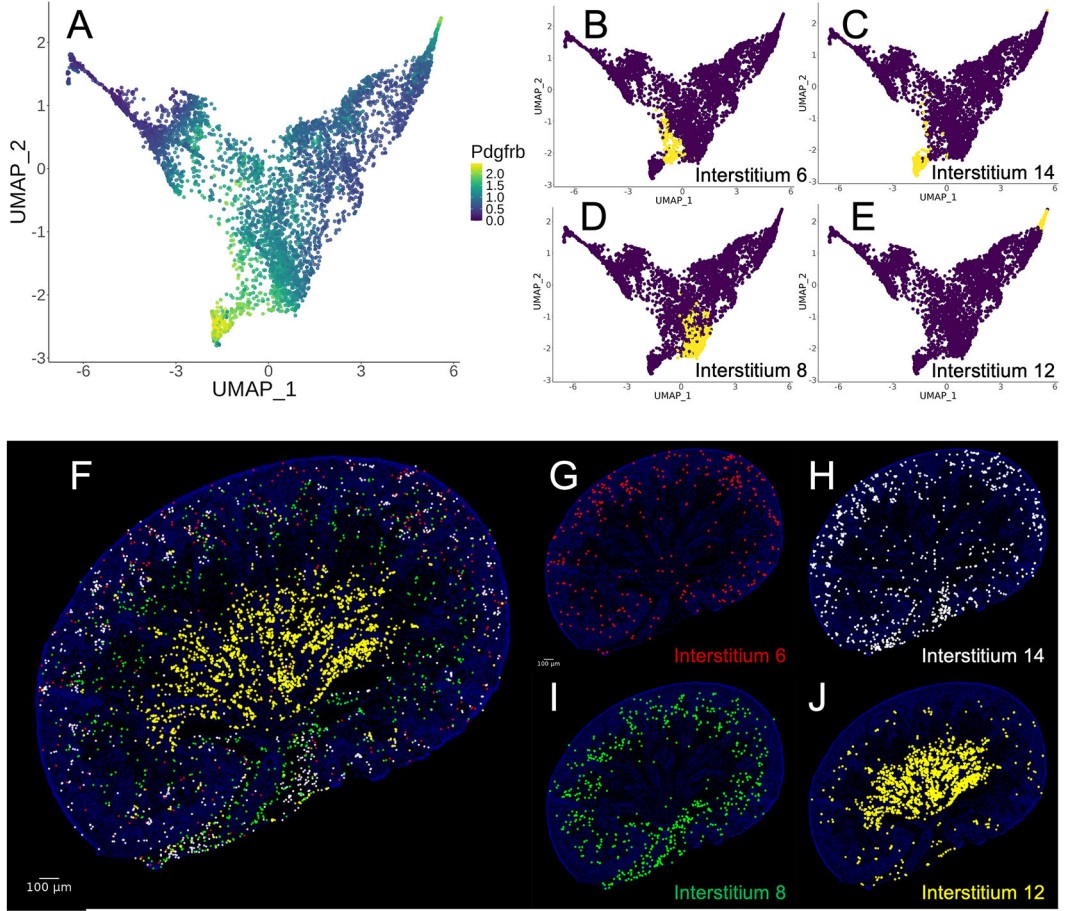

**Fig. 4. Spatial heterogeneity of mural cells.** (A) UMAP of isolated stromal cells colored by expression of the mural cell marker *Pdgfrb*. (B-E) UMAPs indicating four snRNA-seq expression clusters predicted to be marked by *Pdgfrb* expression. (F) Composite prediction of the spatial locations of the four snRNA-seq expression clusters highlighted in B-E (6, red; 14, white; 8, green; 12, yellow). (G-J) Individual predictions of the predicted spatial locations of the four clusters identified in B-E.

confirmed the spatial heterogeneity of the four clusters (Fig. S4L). Additionally, the *in situ* data detected two known mural cell types – mesangial cells and vascular smooth muscle cells – that were not resolved as transcriptionally distinct clusters in the snRNA-seq data (see Discussion).

At E18.5, *Gucy1a1* and *Gucy1a2* were differentially enriched in two of the mural cell clusters discussed above. The Gucy genes encode subunits of soluble guanylate cyclase (sGC), an enzyme activated by nitric oxide that converts GTP to cGMP. This signaling system has been implicated in pericyte maintenance, proliferation and vascular smooth muscle tone regulation (Rees et al., 1989; Stasch et al., 2011; He et al., 2016). The sGC enzyme is a heterodimer consisting of alpha and beta subunits. In our E18.5 and P3 snRNA-seq datasets, *Gucy1b1* was the only beta subunit detected and appeared to encompass the expression domains of both *Gucy1a1* and *Gucy1a2* at E18.5 (Fig. 5A,B, Fig. S5A-C). At P3, *Gucy1a1* and *Gucy1a2* expression remained largely non-overlapping but became more restricted to the juxtamedullary and outer medullary regions, respectively (Fig. 5C,D, Fig. S5D-F). *Gucy1b1* expression overlapped almost entirely with that of *Gucy1a2* and showed minimal overlap with *Gucy1a1*. These observations suggest that distinct isoforms of the alpha and beta subunits are under separate transcriptional control and may contribute to the development/ function of specific pericyte populations and/or distinct developmental processes.

## KSTAT allows spatial mapping of molecular pathways

Our resource can also assign spatial information to predicted transcription factor activity (regulon activity). Using SCENIC (Aibar et al., 2017), we predicted LEF1 regulon activity from snRNA-seq data. This represents a set of genes that are likely co-regulated by LEF1 identified by finding genes expression patterns of which correlate with that of *Lef1* and promoters of which contain its predicted binding motif. The measured activity of the LEF1 regulon is projected onto the E18.5 UMAP in Fig. 6A. Spatial mapping of the LEF1 regulon onto the reference tissue section (Fig. 6B) revealed predicted activity in the medullary stroma, cortical renal vesicles, and ureteric bud. This spatial pattern closely mirrors both the expression of LEF1 (Fig. 6C) and previously demonstrated activity of LEF1 protein in the kidney (Marose et al., 2008; Yu et al., 2002; Drake et al., 2020).

The KSTAT resource also enables cell-level mapping of gene set enrichment for curated gene sets, such as those available in MSigDB (Liberzon et al., 2011). For instance, in Fig. 6D, we mapped the hallmark gene set for G2-to-M progression onto the E18.5 UMAP. Fig. 6E projects the gene set enrichment onto the reference kidney section. Enrichment of the G2-to-M gene set in the kidney cortex aligns with known indicators of cell division. This finding was corroborated by 5-ethynyl-2'-deoxyuridine (EdU) incorporation assays, where a 1.5-h EdU pulse administered to pregnant dams revealed active cell division in the cortical region (Fig. 6F).

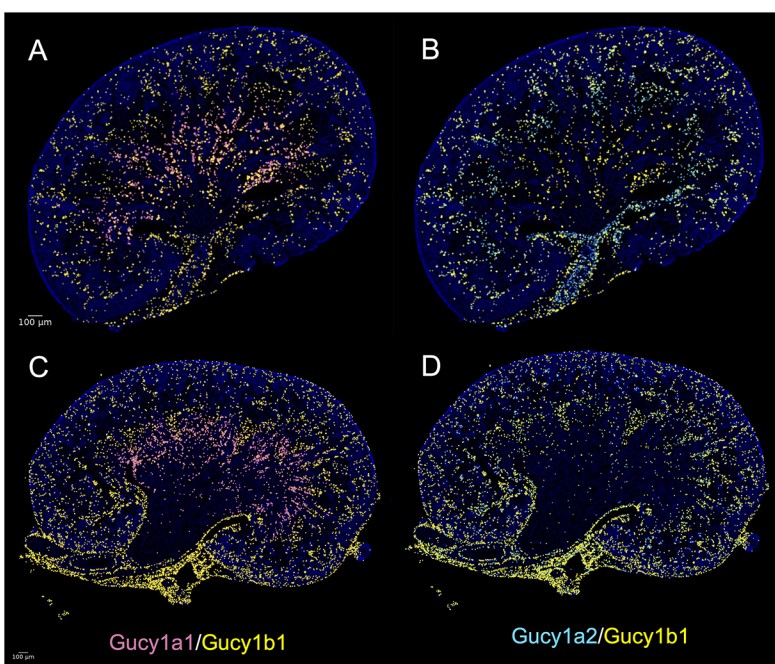

**Fig. 5. Late embryonic and early postnatal heterogeneity of sGC expression.** (A,C) Predicted spatial expression of *Gucy1a1* (pink) and *Gucy1b1* (yellow), the subunits of the canonical sGC heterodimer, at E18.5 (A) and P3 (C). (B,D) Predicted spatial expression of *Gucy1a2* (cyan) and *Gucy1b1* (yellow), the subunits of the alternative sGC heterodimer, at E18.5 (B) and P3 (D).

Next, we evaluated whether metabolic pathways could be spatially mapped in the developing kidney using transcriptomic data. In Fig. 6G, we mapped the Reactome 'metabolism of lipids' gene set onto the E18.5 UMAP. Using AUCell (Aibar et al., 2017) to measure gene set enrichment in individual nuclei, we projected the results onto the reference tissue section (Fig. 6H). The data suggest that lipid metabolism is active in maturing proximal tubules, consistent with recently published findings (Tortelote et al., 2021).

### Spatial information provided by KSTAT improves accuracy of receptor ligand predictions

The early stages of kidney development rely on interactions between mesenchymal and epithelial cells. Our dataset enables the investigation of cell–cell communication by modeling ligand–receptor interactions while incorporating spatial information within the tissue architecture. To test the utility of our dataset in identifying biologically significant interactions, we queried ligand–receptor pairs predicted to be active within the metanephric mesenchyme. Initially, we used CellPhoneDB, an algorithm that does not account for spatial information (Efremova et al., 2020). Among the predicted interactions was the ligand Wnt5a binding to the co-receptor complex Fzd3/Lrp6 (Table S2). Our E18.5 spatial atlas revealed that Wnt5a is expressed in the medullary stroma, while Fzd3/Lrp6 is co-expressed in cortical nephron progenitor cells (Fig. 7A), consistent with previous findings (Yu et al., 2002).

We next used our spatial dataset to refine the analysis by incorporating spatial expression patterns of the interacting proteins. Interestingly, our model deprioritized the Wnt5a/Fzd3 interaction and instead predicted that Wnt9b signals through Fzd3/Lrp5. The spatial dataset confirmed that Wnt9b is expressed throughout the ureteric bud and collecting ducts, while Fzd3/Lrp5 is co-expressed in nephron progenitors (Fig. 7B). Previous genetic work suggests that Wnt9b, and not Wnt5a, signals to nephron progenitors (Carroll et al., 2005; Karner et al., 2011; Pietilä et al., 2016). Nonetheless, we used the data from KSTAT to quantitatively assess the likelihood of these interactions. We utilized GeomLoss (Feydy et al., 2019) to calculate the spatial distances between cells producing the ligands and receptors, reasoning that the likelihood of interaction is inversely related to this distance. Fig. 7C and D depict the predicted receptor-bound ligands, with bound receptors appearing white and saturation levels represented by the opacity of the spots denoting bound receptors: low predicted ligand binding is nearly transparent, while stoichiometric saturation is shown as bright white. Fig. 7C illustrates less-effective binding for the Wnt5a/Fzd3-Lrp6 complex, whereas the Fzd3/Lrp5 receptor complex is almost fully saturated with Wnt9b (Fig. 7D). In Fig. 7A,B, the intensity brightness of red corresponds to the amount of ligand at that location whereas in Fig. 7C,D, the brightness of red indicates the amount of ligand from that location that effectively bound receptor. This analysis is consistent with previously published work and underscores the importance of spatial information in improving predictions of biologically relevant cell–cell communication events.

### DISCUSSION

Recent advances in transcriptomic technologies have enabled the creation of gene expression atlases for various tissues and organs. Traditional approaches, such as antisense mRNA *in situ* hybridization and antibody staining, provide spatial context but are limited by shallow gene coverage and variability between tissue sections, complicating comparisons across genes. Conversely, single-cell transcriptomic datasets offer greater depth, capturing thousands of sequencing reads per cell and profiling thousands of cells simultaneously. However, tissue dissociation in these methods results in a complete loss of spatial context, hindering analyses that rely on understanding cellular relationships within the native tissue architecture. Spatial transcriptomics bridges this gap, enabling simultaneous measurement of multiple transcripts directly in tissue. However, existing platforms are constrained by trade-offs between spatial resolution and transcriptome coverage. Our study addresses these challenges by integrating single-cell transcriptomic data with high-resolution spatial mapping, creating a comprehensive atlas of gene expression in embryonic and postnatal mouse kidneys. This integrative approach provides novel insights into cellular heterogeneity, spatial gene expression, and cell–cell communication in the kidney.

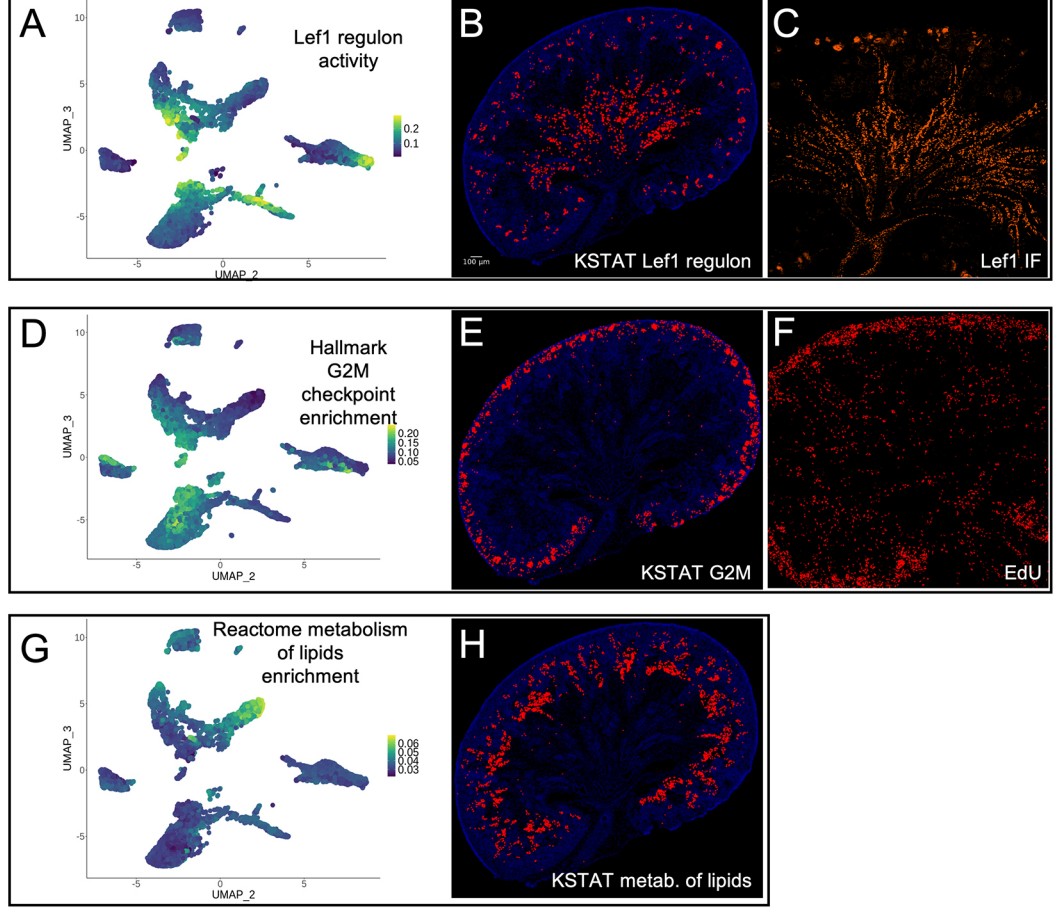

**Fig. 6. Projection of cell properties.** (A) UMAP depicting the results of regulatory network analysis using SCENIC reveals LEF1 regulon activity. (B) The corresponding spatial distribution of highest predicted LEF1 regulon inferred by KSTAT. (C) Immunofluorescence localization of LEF1 in an E18.5 kidney section. (D) Enrichment analysis of MSigDB hallmark genes involved in the G2/M checkpoint projected onto a UMAP. (E) The predicted spatial enrichment of this gene with a threshold applied to show the top percentile. (F) EdU labeling (red) after a 1.5-h pulse in E18.5 mouse kidney. (G,H) Enrichment analysis of lipid metabolism-related genes measured with AUCell (left, UMAP plot). The predicted spatial enrichment of this gene set is shown on the right, with a threshold applied to display the top percentile.

To overcome the limitations of existing technologies, we developed the KSTAT, which combines the depth of single-cell/ nucleus sequencing data with spatial transcriptomics. Using a targeted set of cell type-restricted landmark genes, KSTAT enables spatial assignment of transcriptomic data to (sub)cellular locations within reference E15.5, E18.5 and P3 kidney sections based on the overlap of an individual cell's transcriptome with the spatial transcriptomic data. In a simplified example, we can consider three landmark genes, A, B and C, for which we have ISS data. The expression data for each of those genes is mapped to individual bins, which represent spatial locations on a kidney sample. Every individual bin will show different combinations of expression for each gene (for simplicity, we will say expression is 'on' or 'off' but relative levels of expression can also be measured). The four theoretical bins shown in Fig. S6A show different combinations of expression for each of our three landmark genes. If we then take the transcriptomic data from a single-cell dataset and ask, 'Which spatial pattern does this cell's gene expression most closely match?', this gives us a probability distribution for each cell (Fig. S6B) – a kind of map showing the most likely places in the tissue where the cell could be (Fig. S6C). This integration generates a near-comprehensive spatial atlas of gene expression for a single-cell dataset while also facilitating the correlation of spatial location with advanced analyses, such as cellular annotations, gene set enrichment, and regulon activity. Leveraging KSTAT, we explored mural cell heterogeneity, a feature crucial for understanding disease states and advancing tissue engineering efforts for renal replacement therapies. Our findings revealed previously unrecognized heterogeneity among kidney mural cells at E18.5.

Current methods for visualizing single-cell 'omics data typically rely on dimensionality reduction techniques such as t-distributed stochastic neighbor embedding (tSNE) or UMAP, which position data points (cells) on a two-dimensional map. While distances between data points can indicate relationships between cells, these plots provide no information about the spatial location of cells within the tissue. This low-dimensional representation is useful for some analyses but lacks the spatial context needed for others, such as receptor–ligand interaction studies. Without spatial data, users must rely on prior knowledge or annotations to infer relationships between cells, a process that is both labor intensive and prone to error. KSTAT overcomes this limitation by integrating spatial information, enabling automated, unbiased mapping of transcriptomic data to tissue architecture. This is particularly valuable for researchers new to a model system and for studying pathological tissues, where previously unappreciated cell states may

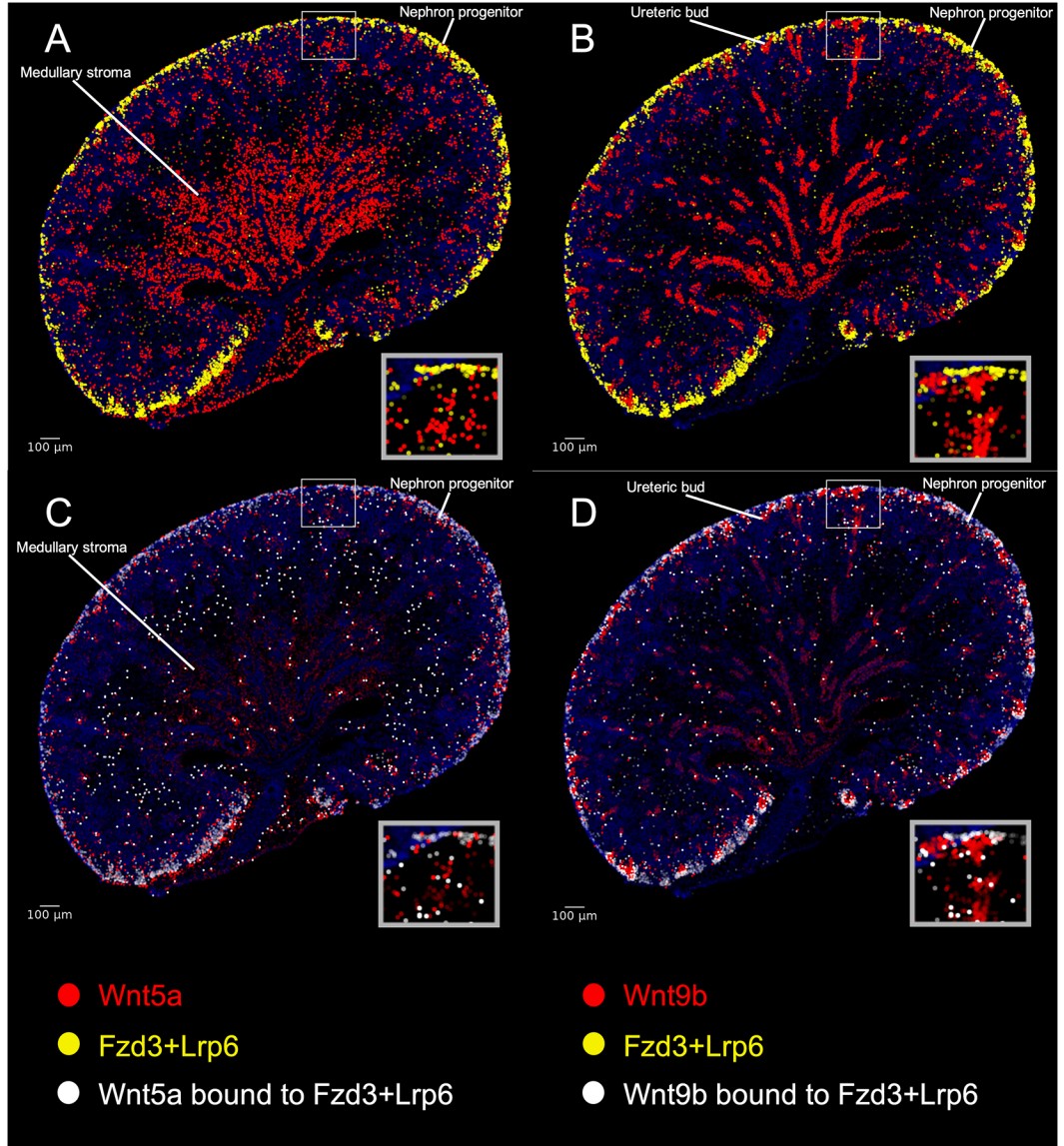

**Fig. 7. Predicting ligand–receptor interactions with KSTAT.** Demonstration of the ability of our model (KSTAT) to predict ligand–receptor interactions in the kidney. (A,B) The predicted spatial distributions of ligand–receptor complexes inferred by KSTAT are shown for two interactions: Wnt5a (red) with co-receptors Fzd3 and Lrp6 (yellow), and Wnt9b (red) with Fzd3 and Lrp5 (yellow). (C,D) The results of optimally transporting the ligands to their receptors. Locations where receptor complexes receive ligand are highlighted in white, with opacity determined by the ratio of ligand to receptor concentration. This visualization illustrates the predicted spatial patterns of ligand–receptor binding.

emerge. By incorporating spatial data into the analysis pipeline, KSTAT expands the utility of single-cell transcriptomics, providing a robust framework for interrogating cellular behavior in both health and disease.

Our validation demonstrates that integrating scRNA-seq or snRNA-seq with limited spatial transcriptomics can accurately predict gene and pathway expression at a near-comprehensive level. However, there are important considerations for researchers aiming to generate similar resources. Chief among these is achieving a high success rate with ISS, which depends on several factors. First, careful selection of landmark genes is crucial. Landmark genes must generate meaningful data, meaning their expression should be detectable but not ubiquitous. Broadly expressed genes must not saturate detection. To address this, we used our previously published algorithm to select landmark genes that fall within this 'sweet spot' of detectability (Chaney et al., 2022).

In our experiment, we successfully detected 88 of the 90 targeted genes, achieving a high success rate. The two undetected genes were likely excluded due to either low expression levels or technical issues during sequencing or visualization. Among the detected genes, four (*H19*, *Fxyd2*, *Acta2* and *Ldhb*) exhibited broad expression patterns. This likely reflects enrichment in specific cell types combined with lower-level expression in others. Re-examination of our snRNA-seq data revealed that these genes had relatively high proportions of cells expressing them above a certain threshold relative to intercluster variance (Fig. S7). Interestingly, genes with more extreme deviations (*S100g*, *Fabp4*, *Wfdc2* and *Mgp*) displayed more spatially restricted expression in the ISS data, suggesting potential limitations in using snRNA-seq to predict saturation in spatial transcriptomics. Despite this, we obtained spatially restricted expression for 86 of the 90 genes, and the saturated signal from these four genes did not significantly impact results due to the redundancy of markers for each cell type. For

instance, the imputed expression of *Acta2* closely matched its known expression pattern.

The success of this technique depends not only on high-quality ISS but also on high-quality transcriptomic data. The greater the depth and quality of the single-cell transcriptomic data, the more comprehensive the resulting atlas will be. Genes or cell types absent from the single-cell dataset cannot be mapped. For instance, the E18.5 kidney dataset used to select landmark genes was enriched for stromal cells but lacked sufficient representation of ureteric and endothelial cells, limiting its capacity to capture pathways active in these compartments. In addition, there were no markers of intercalated cells within our hallmark gene set. Thus, these cells will not be accurately spatially mapped (they will be mapped to general collecting duct). Similarly, since landmark genes were derived from E18.5 data, their representation at E15.5 or P3 may be less robust. Additionally, our single-nucleus sequencing data was generated from whole kidneys without enrichment for specific cell types, potentially under-representing rare populations. This may explain the absence of distinct clusters distinguishing mesangial cells from vascular smooth muscle cells, which were observed through multiplex *in situ* hybridization. Mesangial cells, in particular, are challenging to dissociate from intact glomeruli, complicating their capture. Alternatively, reparameterization of the clustering algorithm could reveal these clusters. Ideally, an exhaustive single-cell dataset with at least five or six landmark genes for every cluster across all time points would maximize accuracy. However, such an approach must balance cost with analytical depth. While other spatial transcriptomics platforms, such as the 10x Genomics Visium platform, offer lower resolution that may hinder certain analyses, our use of the Cartana platform – now incorporated into 10x Genomics Xenium technology – has provided high-resolution data suitable for this work.

Spatial information is particularly important for inferring intercellular communication. Interactions involving membrane-bound ligands and receptors rely on close cellular apposition, while the range of secreted ligand activity depends on the biochemical properties of the extracellular environment. Although computational modeling of signaling relationships based on molecular expression has advanced, incorporating spatial data into such analyses remains a nascent field. Existing spatial transcriptomics platforms often impose limitations on data depth and resolution, constraining their utility. By integrating single-cell and spatial transcriptomics data with computational optimal transport methods, KSTAT overcomes these obstacles. For example, mapping probable Wnt-pathway ligand and receptor locations in the developing kidney demonstrates how spatial data can refine hypotheses about signaling interactions. Our approach assumes that the cellular source of ligand does not influence whether an interaction occurs; instead, receiving cells integrate signals from all sources within their receptive field. This perspective allows us to focus on the transport of ligand species between locations in the reference space, without making assumptions about a cell's capacity for interaction. As such, our method provides a foundation for more sophisticated models that can incorporate additional factors, such as species competition and kinetics of ligand-receptor binding. However, we only considered the ligand mass located within the distance of receptor signal specified by the reach parameter. This integration represents a significant step forward in understanding cell–cell communication, offering insights into potential therapeutic targets and strategies.

A caveat that should be considered is all analyses using KSTAT (or a similar approach) would be done on tissues/cells from mice with an identical genetic background. For instance, if one performed *in situ* transcriptomics on tissues from mice on a 129SvEJ background and scRNA-seq on kidneys from C57Bl6 mice, inaccuracies could arise for genes that are differentially expressed between the two species. Obviously, this caveat exists for all transcriptomic, proteomic or metabolomic datasets. Any data generated using any of these techniques needs to be validated by the user.

By integrating spatial data with single-cell transcriptomics, KSTAT enables researchers to generate specific hypotheses about tissue development. However, beyond kidney development, this study provides a roadmap for how to generate additional expression atlases. In pathological contexts, novel cell states often emerge and influence disease progression. Our approach allows precise mapping of disease-associated transcriptional programs or cell states within tissues, providing a foundation for identifying spatially localized signaling pathways. This knowledge can inform the development of targeted therapeutic strategies by pinpointing communication networks that regulate key transcriptional programs. Ultimately, these types of resources will open up new avenues for understanding and modulating development, maintenance, regeneration and disease, offering powerful tools for advancing precision medicine.

## MATERIALS AND METHODS

### Isolation of nuclei and sequencing
All animals were housed, maintained and used according to National Institutes of Health (NIH) and Institutional Animal Care and Use Committees (IACUC) approved protocols at the University of Texas Southwestern Medical Center (OLAW Assurance Number D16-00296). To isolate nuclei, we modified a protocol from Ben Humphrey's lab (Wilson et al., 2019). Kidneys from E18.5 and P3 kidneys of mice on a mixed genetic background were dissected in cold PBS without calcium or magnesium, snap-frozen, and then thawed on ice for pooling. We minced the kidneys using razorblades and homogenized them with a Dounce homogenizer in Nuclei EZ Lysis buffer (Sigma-Aldrich) supplemented with protease inhibitor (Roche) and RNase inhibitor (Promega and Life Technologies). The homogenate was filtered through a 40-μm cell strainer (pluriSelect) and centrifuged at 500 $g$ for 5 min at 4°C. The pellet was resuspended, washed, and filtered again through a 5-μm cell strainer (pluriSelect).

The resulting nuclear suspensions were submitted to UTSW Next Generation Sequencing Core for library preparation and sequencing using the Chromium Single Cell 3′ Prime Gene Expression kit (10x Genomics) on an Illumina NextSeq 2000.

### sc/snRNA-seq preprocessing
Cells (nuclei) were called from empty droplets by testing for deviation of the expression profile for each cell from the ambient RNA pool (Lun et al., 2018). Cells (nuclei) with low numbers of detected genes or library sizes, as defined as a deviation greater than three median absolute deviations below the median, were excluded from further analysis. Similarly, cells (nuclei) with large mitochondrial proportions, i.e. more than three median absolute deviations above the median, were removed. However, in the case of stripped nuclei, we enforced a minimum difference of 0.5% between the threshold for exclusion and the median mitochondrial transcript count. Cells (nuclei) were pre-clustered; a deconvolution method was applied to compute size factors for all cells (Lun et al., 2016) and normalized log-expression values were calculated.

### Landmark selection
Cells were clustered by building a shared nearest neighbor graph, which represents cells as nodes connected by edges based on their similarity (Xu and Su, 2015). The Walktrap algorithm (Pons and Latapy, 2006) was then applied to identify clusters of cells with similar characteristics. After coarsely annotating the cells, stromal cells were further subclustered. In total, 52 clusters were found, 17 of which were stroma. To identify genes that specifically mark populations of sequenced cells, we employed the EIGEN algorithm, which leverages the 'wisdom of the crowds' principle. This involves measuring pairwise differential expression between each pair of clusters using multiple statistical methods (pairwise *t*-test, pairwise Binom, and pairwise Wilcox from the scran Bioconductor package, as well as the

zlm method from the MAST Bioconductor package). A manual curation step was performed to ensure both specific expression and coverage of all cell states represented in the sequencing data.

## ISS

Tissue sections were collected and pre-processed according to established protocols provided by Cartana, a platform specializing in direct RNA targeted ISS. All images were captured using a 20× objective with an NA of 1.0. Localization precision for individual transcripts was <30 nM. The resolution of the assigned pixels was 200 nM. Cartana has since been acquired by 10x Genomics. Of the genes selected for analysis, data could not be acquired for *Amd1* and *H1f5* due to technical limitations.

## Preprocessing

Reads were provided by Cartana as fractional coordinates of segmented spots in a 16,619×14,810-pixel reference space, which represents the spatial distribution of gene expression within the tissue section. The data were processed as a binary array, where each element $(x, y, l)$ indicates whether landmark gene $l$ is expressed at location $(x, y)$. To reduce computational complexity and sparsity, we aggregated data from 16×16-pixel tiles, effectively reducing the height and width to 926 and 1039 bins, respectively. This step allowed us to identify 32,157 unique bit vectors of landmark expression, with different combinations occurring at varying frequencies.

## Probability inference

We analyzed snRNA-seq from E15.5, 18.5 and P3 kidneys that were collected separately from the scRNA-seq data used for landmark identification as above. UMAPs for these datasets annotated by cell type are shown in Fig. S8A, C and E, respectively. Heatmaps show mean expression of genes representing the various cell types. Two of the landmark genes chosen had no reads in the E18.5 snRNA-seq data, which led to their exclusion from further analysis due to insufficient information.

Following the methods of Satija et al. (2015), we modeled the expression of each landmark gene in the snRNA-seq dataset as a mixture of normal distributions. However, our approach differed from theirs in terms of data imputation and mixture component inference. We began by representing the marginal distribution of expression for each landmark gene in the snRNA-seq dataset as a two-component mixture of normal distributions. Droplet capture sequencing results are represented by integer counts, demonstrate sparsity and have been observed to be zero-inflated, all of which are barriers to analysis with Gaussian models. These issues are partially addressed during normalization. We observed that imputation on normalized data using Markov Affinity-based Graph Imputation of Cells (MAGIC) (Van Dijk et al., 2018) smoothed the data and made it much more amenable to assumption of normality (see Fig. S9).

For each of the 88 landmark genes analyzed, we fit Gaussian mixtures with k components for k in {2, 3, …, 8} and selected the mixture with maximum likelihood. The component with the highest mean was chosen to represent cells that expressed the landmark gene, the 'on' population, and the remaining components were combined to represent cells that did not express the gene, the 'off' population. At the completion of this stage, we possessed estimates of the mean and covariance in 'on' and 'off' populations for each landmark gene.

Next, we estimated multivariate normal distributions representing the probability of a cell occupying each bin as a function of that cell's expression of the landmark genes and the measured landmark signature for that bin. For each landmark, we chose the parameters for its marginal distribution from those inferred above according to whether there was signal from the landmark measured at that location. This readily yielded an estimate of the one-dimensional mean vector. The covariance was approximated by a matrix with the variances calculated for each landmark's selected component along the diagonal.

With the inferred distributions corresponding to each bin with a unique landmark signature in hand, calculation of the likelihood that a cell was located at a specific bin was reduced to an embarrassingly parallel computation of multivariate density. The likelihood of each transcriptome occupying the bin was then calculated using the 'dmvnorm' function from the 'mixtools' (Benaglia et al., 2009) R package.

Ultimately, one obtains an $(N×B)$-dimensional tensor encoding an estimate of the likelihood of each of the $N$ cells being found at each of the B unique bins in the $H×W$ reference space. To permit further computation, normalization to yield proper probability distributions for each unique bin was required. Rather than application of 'softmax' or sum-normalization, which would result in full support over the population of cells, we employed 'sparsegen-lin' (Laha et al., 2018), an adaptation of the 'sparsemax' transformation, (Martins and Astudillo, 2016), which allows control of the degree of sparsity. The inferred probabilities were broadcast to all bins with the same landmark expression signature yielding an $N_c×N_b$ dimensional matrix $P$, where $N_c$ is the number of transcriptomes (cells or nuclei) and $N_b$ is the number of bins.

## Projection of cell features

With the above inferred probability distributions in hand, we were able to proceed to the projection of various properties of the cells onto the reference space. This involved calculating an expectation over each bin's distribution, which enabled us to quantify specific features of interest. For instance, the expected expression of gene $g$ at pixel $i$ is given by:

$$x_{g_i} = e_g \cdot P_{.,i}.$$

Since the probability distribution is inferred bin-wise, transitions between adjacent pixels are inherently abrupt. This leads to high-frequency fluctuations in the signal, which can be problematic. To address this, we reduced noise by applying a Gaussian filter to the expectation suppressing high frequencies while simultaneously minimizing spatial spread. The signal could then be visualized as the total expectation (Fig. 2, second row) or a threshold applied to yield a virtual *in situ* hybridization image as depicted for three different thresholds in the lower part of Fig. 2.

In addition to projecting continuous features, it is also possible to project categorical properties of cells using a similar approach. This involves encoding whether a given cell belongs to a particular category, which can be represented as a binary-valued indicator function supported by the cells. As an example, transfer of the annotation of cells in the sequencing dataset as those forming the proximal tubule were encoded as a binary vector with one indicating the presence of the annotation and 0 otherwise. On right-multiplying by the cells×bins probability matrix, one obtains the expected value of the indicator function at each pixel, which is proportional to the likelihood that an annotated cell corresponds to that location. This calculated expression can be used for further downstream analysis or binarized by thresholding for visualization.

## Automatic thresholding

We developed an automatic thresholding procedure based on the observation that the inferred signal, when ordered in ascending order, typically assumes an inverted sigmoid shape. To identify the elbow point of this curve, we implemented a method that requires a user-defined hyperparameter k. First, we sorted the unique non-zero expected values in ascending order. Then, we fit two linear models: one to the k smallest values starting from the median point, and another to the k largest values. The intersection point of these two lines was calculated, and the corresponding expected value at this point was identified as the threshold. The results of applying this thresholding procedure to the landmark *Nphs1* are illustrated in Fig. S10.

## Assignment of probability of marking by *Pdgfrb*

To determine that a cluster was marked by *Pdgfrb* expression, we conducted the following analysis. The data included the imputed expression of *Pdgfrb* in all of the $N$ cells and the assignment of each cell to one of $J$ clusters. We assumed the mean expression of *Pdgfrb* in each cluster, $\mu_k$, to be normally distributed, the unknown threshold that separated clusters marked by expression of *Pdgfrb* from those that do not, $T$, was normally distributed and the slope controlling the sharpness of the decision boundary, alpha, was Cauchy distributed:

$$\mu_i \sim \mathcal{N}(0.3699511, 0.4939401), \quad \text{for } i = 1, \ldots, N,$$
$$\sigma_i \sim \text{Cauchy}(0.07308471, 0.1083554), \quad \text{for } i = 1, \ldots, N,$$
$$T \sim \mathcal{N}(1, 0.5),$$
$$\alpha \sim \text{Cauchy}(0, 1).$$

The mean and standard deviation to parameterize the prior distribution of cluster means and the median and median absolute deviation to parameterize the prior distribution of cluster standard deviations were estimated from the expression of all genes in all cells. Uninformative priors were used for $T$ and $\alpha$ as there was no information with which to refine the assignments.

The likelihood of the expression of *Pdgfrb* in cell $j$ was modeled as:

$$y_j \sim \mathcal{N}(\mu_{\text{cluster\_idx}_j}, \sigma_{\text{cluster\_idx}_j}), \quad \text{for } j = 1, \ldots, J.$$

Finally, the making probability for each cluster was defined as:

$$\text{marked\_prob}_i = \text{inv\_logit}(\alpha \cdot (\mu_i - T)), \quad \text{for } i = 1, \ldots, N$$

where

$$\text{inv\_logit}(x) = \frac{1}{1 + e^{-x}}.$$

The posterior distribution of the marking probability for each cluster was sampled 10,000 times and the estimates of the mean and 89% credible intervals are displayed in Fig. S11.

### Performance evaluation

To quantitatively assess the performance of KSTAT, we measured the reconstruction error between the measured landmark distribution of signal and that inferred using KSTAT. To reduce computational complexity and normalize the comparisons, once we inferred expression for a given landmark gene, we calculated a threshold so that the binarized signal would match the cardinality of the measured landmark signal. We used the Sinkhorn divergence to quantify this distance between the distributions. While this allowed us to quantify the ability of KSTAT to faithfully reconstruct the landmark signal with the fully trained model, it did not inform the likelihood of successful inference on data not previously seen by KSTAT. Thus, we also performed LOOCV. For each landmark, we inferred a model from all remaining landmarks and used this model to infer the left-out landmark's distribution and then measured the error between the inferred and measured values as above. We approximated the probability of obtaining the distribution inferred by KSTAT with a bootstrap distribution derived separately for each landmark. In each case, 10,000 bins were sampled from a uniform distribution supported by the two-dimensional reference space and the Sinkhorn divergence between the sampled and measured distribution was calculated. The size of the sample for each landmark was chosen to match the cardinality of the landmark signal. The lower tail mass of this distribution determined by either the reconstruction or LOOCV divergences provided an estimate of the likelihood of observing the divergence by chance. These quantitative evaluation methods provided a comprehensive assessment of KSTAT's performance and robustness.

### Cell–cell communication inference

We investigated cell–cell communication from the perspective of receiving cells, focusing on the distribution of each component of curated receptor complexes in the population of these cells. Our analysis was restricted to ligand–receptor interactions curated in the CellPhoneDB v.5.0 database. To ensure accurate calculations, we imposed stoichiometric constraints by taking the minimum expression across receptor components for each cell.

Next, we integrated the abundances of all ligand and receptor components with the debiased Sinkhorn divergence between the ligand and receptor distributions, which serves as a low-cost approximation to the earth mover's (Wasserstein) distance. To model the spatial constraints of ligand diffusion, we used the reach parameter of the geomloss Sinkhorn SampleLoss instance to account for the limited ability of a ligand species to diffuse through space.

The code for this analysis as well as a detailed tutorial on how to regenerate the data are available at https://github.com/cpchaney/kstat.

### Kidney sample preparation, *in situ* hybridization, immunostaining and EdU labeling assays

Embryonic tissue was fixed in 4% paraformaldehyde and processed for immunostaining or *in situ* hybridization as previously described (Drake et al., 2020). cDNAs for antisense probes were purchased from OpenBiosytems/Invitrogen. The Lef1 antibody was purchased from Proteintech (2230S, dilution 1:200). To detect cell proliferation, 25 μg/g body weight

5-ethynyl-2'-deoxyuridine (EdU) was injected intraperitoneally into pregnant dams at E18.5 and embryos harvested after 1.5 h of labeling. Kidneys were dissected, fixed and sectioned as described above. The Click-It EdU kit (Thermo Fisher Scientific, C10337) was used to detect EdU$^+$ cells and sections co-stained with tdTomato and LTL. Images were captured with Nikon A1R confocal microscope using a 40× oil objective and whole-kidney images were created with automatic tiling of the tissue with Nikon Elements software.

### Declaration of generative AI and AI-assisted technologies

During the preparation of this work, the authors used llama3 in order to improve readability and language of the manuscript. After using this tool/service, the authors reviewed and edited the content as needed and take full responsibility for the content of the published article.

#### Acknowledgements
We would like to thank Dr Morgane Rouault, Dr Courtney Karner, Dr Robert Tower and members of the Carroll, Cleaver and Marciano laboratories for reading and providing comments on this study and manuscript. This work was assisted by the UTSW Next-Gen Sequencing Core.

#### Competing interests
The authors declare no competing or financial interests.

#### Author contributions
Conceptualization: C.P.C., T.J.C.; Data curation: C.P.C., A.N.F., E.D.G., J.N.W., P.M.L.; Formal analysis: C.P.C.; Funding acquisition: O.C., D.K.M., T.J.C.; Investigation: C.P.C., A.N.F., E.D.G., J.N.W., P.M.L.; Methodology: C.P.C., T.J.C.; Project administration: T.J.C.; Resources: O.C., D.K.M., T.J.C.; Supervision: O.C., D.K.M., T.J.C.; Validation: C.P.C., T.J.C.; Visualization: C.P.C., T.J.C.; Writing – original draft: C.P.C.; Writing – review & editing: T.J.C.

#### Funding
Financial support for this work was provided by grants from the National Institutes of Health National Institute of Diabetes and Digestive and Kidney Diseases (NIDDK) (RC2DK125960 to T.J.C., D.K.M. and O.C.; R01DK127634 to T.J.C.; RO1DK124393 to O.C.; and UC2DK126021 to D.K.M. and C.P.C.) and from the NIDDK Diabetic Complications Consortium (DiaComp, www.diacomp.org: DK076169 to T.J.C. and C.P.C). Open Access funding provided by the University of Texas Southwestern Medical Center. Deposited in PMC for immediate release.

#### Data and resource availability
snRNA-seq data have been deposited at Gene Expression Omnibus under accession number GSE284496. All original code has been deposited at Zenodo (https://doi.org/10.5281/zenodo.17781737). STAT software is available at https://github.com/cpchaney/kstat. All other relevant data and details of resources can be found within the article and its supplementary information.

#### Peer review history
The peer review history is available online at https://journals.biologists.com/dev/lookup/doi/10.1242/dev.205003.reviewer-comments.pdf

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
