## [Peer Review File · Development (Cambridge, England)]

Integration of spatial and single-nucleus transcriptomics to map gene expression in the developing mouse kidney

Christopher Chaney, Alexandria N. Fusco, Elyse D. Grilli, Jane N. Warshaw, Peter M. Luo, Ondine Cleaver, Denise K. Marciano and Thomas J. Carroll

DOI: 10.1242/dev.205003

Editor: Liz Robertson

Review timeline

Original submission:	3 June 2025
Editorial decision:	9 July 2025
First revision received:	14 October 2025
Accepted:	10 November 2025

Original submission

First decision letter

MS ID#: dev.205003

MS TITLE: Integration of spatial and single nucleus transcriptomics to map gene expression in the developing mouse kidney

AUTHORS: Christopher Chaney, Alexandria N. Fusco, Elyse D. Grilli, Jane N. Warshaw, Peter M. Luo, Ondine Cleaver, Denise K. Marciano and Thomas J. Carroll

Dear Tom,

I have now received all the referees reports on the above manuscript, and have reached a decision. The referees' comments are appended below, or you can access them online: please go to .

The overall evaluation is positive and we would like to publish a revised manuscript in Development, provided that the referees' comments can be satisfactorily addressed. Please attend to all of the reviewers' comments in your revised manuscript and detail them in your point-by-point response. If you do not agree with any of their criticisms or suggestions explain clearly why this is so. If it would be helpful, you are welcome to contact us to discuss your revision in greater detail. Please send us a point-by-point response indicating your plans for addressing the referees' comments, and we will look over this and provide further guidance.

Reviewer 1

SUMMARY OF THE ADVANCE MADE IN THIS PAPER AND ITS POTENTIAL SIGNIFICANCE TO THE FIELD

This work describes development of an analysis package to map single cell transcriptomic data spatially onto an image of a kidney section. The technique is based on the identification and validation of around 90 anchor genes in broad categories such as "endothelium", "epithelium" etc. Validation using ISH is convincing and computational tests of prediction reveal high accuracy. Examples are provided for the use of the technology, including co-localization of gene expression to predict cellular composition of structures within tissue, regulon analysis, and ligand-receptor prediction. A series of developmental stages are presented, although data looks more robust from E18.5 onwards. In all, this provides a useful and timely tool that can be very helpful for

investigation of normal kidney development, and I think it will be of great interest to readers of Development. Two areas that I think need to be expanded to manage expectations of a developmental biology audience: 1) What are the limitations/obstacles to using this for the study of genetic/toxicological/pharmacological perturbations of kidney development? 2) How do I use this interesting new tool to analyze my data?

SUGGESTIONS TO AUTHORS

1. Order the ISH images alphabetically - it is difficult to cross reference otherwise.
2. Check in-text reference to 6H.
3. The authors should demonstrate how much magnification can be expected; for example overlap in figure 7 is difficult to assess with the magnifications provided. An inset in each of the whole-organ panels with a magnification of the relevant field would be very helpful.
4. Scale bars would be a helpful addition so that investigators easily can cross-reference the data from KSTAT with immunostaining and other methods.
5. Many investigators will be enthusiastic to use this technology to investigate genetic or pharmacological perturbations. In the discussion, it would be helpful to provide some guidance on the limitations of the technology in this regard and point out which validations would need to be done to understand the robustness of the anchor gene-set across conditions.
6. To manage expectations, it would also be helpful to describe any plans to provide access to this analysis platform in the discussion, or if the plan is for individual labs to implement the software themselves. If the latter, a description of how to do that in an accompanying protocols/methods paper would be helpful. This is a very useful tool for the field, so a clear path to implementation is a priority.

Reviewer 2

SUMMARY OF THE ADVANCE MADE IN THIS PAPER AND ITS POTENTIAL SIGNIFICANCE TO THE FIELD

This resource article by Chaney, Carroll, and colleagues describes a novel spatial atlas of gene expression in the developing kidney. The authors integrated single cell/nucleus RNA-seq data with high-resolution spatial gene expression arrays to create the computational tool (KSTAT) that imputes transcriptome-wide spatial expression patterns across multiple developmental timepoints in the form of a virtual 'ISH' image. Appropriate confirmational validation was done comparing predicted expression to real ISH assays. The authors further demonstrated the tractability of the software by showing the ability to spatially map metadata information, regulon and/or gene set scoring, and pathway activities. And they also integrated methods for prediction of cell-cell communication that uses both ligand/receptor expression and distance between cells that are putatively signaling, with this approach showing that reliance on only scRNA-seq is likely to predict non-meaningful signaling interactions between cells that are spatially isolated.

Overall, this resource is a valuable and unique contribution to the kidney developmental biology field. To my knowledge, there is no comparable tool existing in the literature and current transcriptome-wide spatial sequencing technologies cannot achieve the degree of spatial resolution offered by this integrated KSTAT approach. Further, I have not seen the authors' approach of combining scRNA-seq with targeted spatial sequencing reported in development of other organs beyond the kidney. Therefore, I predict this paper will be of high interest and utility to not only the kidney development community, but also extending to the broader readership of Development.

SUGGESTIONS TO AUTHORS

There are no major holes or flaws with the paper in its current form, and all the authors' conclusions are sufficiently supported by their data. The manuscript reads as a bit technical/computational for a Development paper, especially the Results section, but I suppose this is to be expected given the nature of the work. As a non-technical expert, I am still able to follow the logic and understand the conclusions even if not truly comprehending the methods. I have only a few minor suggestions that might improve the manuscript.

Minor recommendations

1. Most of the examples and analyses shown in the paper focus on the developing renal interstitium, which is a focus of the authors' lab. It might be useful to add some anecdotal data that also shows the projection of epithelial cell types or genes, which would likely be of interest to many readers and potential users of this Resource. For example, showing the localization of several nephron segments based on their clustering in the scRNA-seq data, analogous to what is shown in Fig. 4F-J.
2. Also, the UMAPs of the three scRNA-seq datasets with cluster annotations should be included in the supplement so that readers can understand all the cell types they should expect to be able to interrogate using KSTAT.
3. Figure 5. Improved annotation of the figure to specify which images are E18.5 vs P3 would be helpful. This information is also not present in the figure legend, which is minimal.
4. Labeling the panels (6C = LEF IF staining; 6F = EdU incorporation) in Figure 6 would enhance the reader's ability to interpret the figure. In Figure 7, directly labelling on the figure that white = bound receptors and yellow is unbound would be nice.
5. Subheadings in the Results section would be helpful.

Reviewer 3

SUMMARY OF THE ADVANCE MADE IN THIS PAPER AND ITS POTENTIAL SIGNIFICANCE TO THE FIELD

The manuscript by Cheney et al integrates single-nucleus RNA sequencing with high-resolution spatial transcriptomics with the aim of generating a comprehensive spatial and transcriptional atlas of embryonic and postnatal mouse kidneys. The authors introduce the Kidney Spatial Transcriptome Analysis Tool (KSTAT), which enables spatial mapping of gene expression, prediction of cell-cell communication, and pathway activity analysis, among other features. With this tool, the authors identify previously unappreciated heterogeneity among embryonic kidney pericytes. The authors generated this resource due to limitations of traditional transcriptomic approaches that lack spatial resolution and/or significant depth. Overall, this appears to be an effective approach to mapping single cell datasets back onto the kidney to generate predicted expression patterns and gain biological insights. While this approach could be useful for the broader developmental biology community, taking advantage of the numerous publicly available single cell/nucleus datasets, the resource is somewhat narrowly focused on the kidney and the author's interests. If the authors are unable to expand the analysis to include non-Interstitial biased in situ targets or other tissues, then it would still be useful to "market" this as a potentially more broadly applicable approach in the writing. Other notable concerns are included below.

SUGGESTIONS TO AUTHORS

1) The paper states that the authors developed a tool (Kidney Spatial Transcriptome Analysis Tool) and a virtual atlas. The impression given is that this might be a web resource that any user could utilize to look up their favorite gene, map a pathway, etc. and receive an output/image. What is the actual tool/resource-is it the method? Using the term "tool" feels like the reader should be provided something-some resource that is the result of the research performed in this manuscript. If there is no actual tool that can be identified, such as an online tool (like the Kidney Interactive Transcriptomics website) or code or GitHub that encompasses the entire pipeline (like a plug n play resource), the reviewer would recommend not naming this as a "tool" or resource but as a methodology they developed that could be applied to the kidney or other tissues. In that case, a step-by-step protocol that describes how each reader can make their own "resource" like the KSTAT would be of more significant utility.

Edit: This was written before using the "find" function on the manuscript and finding the GitHub link a last effort, buried in the Resource table near the end. This appears to be the resource generated but is not clear at all from reading the manuscript.

- 2) The Results seem somewhat of a rapid fire of what the resource can do, with short 2-3 sentence descriptions of what was performed. There is no significant depth to these with generally 1 example being presented each (ie G2 to M analysis, the Lef regulon, the metabolism of lipids pathway). Each of these examples then nicely matches what is already known. While validation is important, showing one example is not very robust. Showing that the resource could provide novel biological insights, ie mapping a pathway or similar that is not well described in the kidney and then validating its significance through genetic or kidney culture experiments would significantly enhance the utility of this resource.
- 3) The authors state that their methodology/dataset "offers unprecedented resolution and depth." If this is the case, the authors should provide a direct comparison to other spatial transcriptomic analyses-how much more resolution and depth does it provide? Numbers would help support their claim. How does their methodology compare to a spatial technology such as Stereo-seq from Complete Genomics which can resolve single cells?
- 4) The authors state that "This approach enabled us to map transcriptional data from each cell or nucleus onto a spatial framework, effectively generating a virtual in situ hybridization image for every gene expressed in the datasets." To this reviewer it is unclear how every cell can be mapped, or what the coverage is. Perhaps in a single nucleus/cell sequencing dataset there are 3,000 interstitial cells that are sequenced. How many interstitial cells are then on an E18.5 section? Less than 3,000 or more than 3,000? If less, then how is every cell mapped and if more then are there cells on the section that don't have a correlating single nucleus/cell? Maybe there is some misunderstanding on the reviewer's behalf, but I think this should be clarified further in the text.
- 5) Fig 3A should include higher mag views of the in situ and predicted patterns so the reader can more accurately see and assess the comparison. The current view is too low resolution even upon zooming to assess in what cells/structures these are expressed or predicted to be expressed.
- 6) Results say E18.5 and P3 single nucleus data was generated but the Methods only describe E18.5. It says that kidneys were snap frozen for genotyping. What genotyping needed to be performed on these samples? There is no information provided on the strain of mice that were used. This needs to be added. Also, if the data generated is on a different background strain of mice than the publicly available datasets, does this matter at all for the tool or interpretation? Does this factor need to be considered?
- 7) Text states that Gucy1b1 was the only beta subunit detected at E18.5 and P3. But later it is stated that Gucylb2 overlaps with Gucyla2. Were both detected by the spatial profiling but not single cell data? This is not clear.
- 8) What is regulon activity for predicting transcription factor activity? This should be more clearly explained how this is done for readers who are not familiar.
- 9) There is reference to the "statistical model used to predict likelihood of a cell's location." Is this a previously published method or did the authors generate this? It should be described better how this was developed.
- 10) In Figure 1, why are Cited1 and Six2, which predominantly mark mesenchymal nephron progenitors classified as epithelial? This classification should be clarified so as to include epithelial progenitors or similar.
- 11) A schematic of the approach/technique should be added to help readers/researchers better understand what was necessary to generate the resource/tool-what does the pipeline look like? It took several readings to begin to understand all the experiments and analyses performed.
- 12) More specifics on when genes/things don't map properly should be provided. For example, for the 84 of 88 genes that had accurate spatial predictions, how off or wrong were the other genes?
- 13) Add subsections to the Results to help readers follow and break up sections
- 14) Figure legends for the main figures are very minimal and should be expanded. For example, Figure 5 legend is only a title.

First revision

Author response to reviewers' comments

We want to extend our appreciation to the reviewers for taking the time to carefully read and comment on our manuscript. Although in principle, we agree with all of the comments, we do not

feel that we are able or need to respond to each one within the context of this manuscript. We hope that reviewers agree that we have responded to the majority of their comments sufficiently and that they have improved the manuscript significantly and that it is now appropriate for publication at *Development*. Below, we have included the original reviews. Our response to each is included in green.

Reviewer 1: SUMMARY OF THE ADVANCE MADE IN THIS PAPER AND ITS POTENTIAL SIGNIFICANCE TO THE FIELD

This work describes development of an analysis package to map single cell transcriptomic data spatially onto an image of a kidney section. The technique is based on the identification and validation of around 90 anchor genes in broad categories such as "endothelium", "epithelium" etc. Validation using ISH is convincing and computational tests of prediction reveal high accuracy. Examples are provided for the use of the technology, including co-localization of gene expression to predict cellular composition of structures within tissue, regulon analysis, and ligand-receptor prediction. A series of developmental stages are presented, although data looks more robust from E18.5 onwards. In all, this provides a useful and timely tool that can be very helpful for investigation of normal kidney development, and I think it will be of great interest to readers of *Development*. Two areas that I think need to be expanded to manage expectations of a developmental biology audience: 1) What are the limitations/obstacles to using this for the study of genetic/toxicological/pharmacological perturbations of kidney development? 2) How do I use this interesting new tool to analyze my data?

SUGGESTIONS TO AUTHORS

1. Order the ISH images alphabetically - it is difficult to cross reference otherwise. This has been done.
2. Check in-text reference to 6H. An updated figure has been included that matches the text citations.
3. The authors should demonstrate how much magnification can be expected; for example overlap in figure 7 is difficult to assess with the magnifications provided. An inset in each of the whole-organ panels with a magnification of the relevant field would be very helpful. The image capture for the in situ sequencing was all done at a single magnification with a 20x objective with an NA of 1.0 but images are captured at very high resolution (localization precision for individual transcripts is <30nm). The "predicted" images can be electronically magnified as much as one wants but the resolution of the assigned pixels is 200nm. We have added text to the methods to clarify this. In addition, we have added higher magnification insets in Figure 7 to show relevant fields, with bounding boxes indicating the inset area.
4. Scale bars would be a helpful addition so that investigators easily can cross-reference the data from KSTAT with immunostaining and other methods. This has been done.
5. Many investigators will be enthusiastic to use this technology to investigate genetic or pharmacological perturbations. In the discussion, it would be helpful to provide some guidance on the limitations of the technology in this regard and point out which validations would need to be done to understand the robustness of the anchor gene-set across conditions. The current atlas is only useful for assessing wildtype kidneys. The tool can be used to model gene expression/pathway analysis in genetic, toxicological, or pharmacologic perturbations relative to reference systems. We have provided a detailed guidance for how to create a similar atlas in GitHub.
6. To manage expectations, it would also be helpful to describe any plans to provide access to this analysis platform in the discussion, or if the plan is for individual labs to implement the software themselves. If the latter, a description of how to do that in an accompanying protocols/methods paper would be helpful. This is a very useful tool for the field, so a clear path to implementation is a priority. This situation is in flux. The original agreement was that the NIH would host this dataset through the Gudmap website. Unfortunately, due to funding cuts, the NIH has just announced that the Gudmap website will be taken down. The cost to support a website that is open access is currently prohibitive for our lab (over \$200k/year). We are still seeking solutions to this issue. However, as noted by reviewer 3, we do provide a link to the data and code on Github and we have now included a detailed description of how to recreate our atlas in individual labs. In the revised manuscript, we will:

- A. Expand the Discussion to explicitly state that the full analysis pipeline, including all code and data, is available on GitHub.
- B. Develop a **step-by-step tutorial** on how to reproduce the results, including a worked example, and link to it prominently in both the main text and supplementary materials.
- C. Include the **Shiny app code** and instructions for adaptation to user data.

Reviewer 2: SUMMARY OF THE ADVANCE MADE IN THIS PAPER AND ITS POTENTIAL SIGNIFICANCE TO THE FIELD

This resource article by Chaney, Carroll, and colleagues describes a novel spatial atlas of gene expression in the developing kidney. The authors integrated single cell/nucleus RNA-seq data with high-resolution spatial gene expression arrays to create the computational tool (KSTAT) that imputes transcriptome-wide spatial expression patterns across multiple developmental timepoints in the form of a virtual 'ISH' image. Appropriate confirmational validation was done comparing predicted expression to real ISH assays. The authors further demonstrated the tractability of the software by showing the ability to spatially map metadata information, regulon and/or gene set scoring, and pathway activities. And they also integrated methods for prediction of cell-cell communication that uses both ligand/receptor expression and distance between cells that are putatively signaling, with this approach showing that reliance on only scRNA-seq is likely to predict non-meaningful signaling interactions between cells that are spatially isolated.

Overall, this resource is a valuable and unique contribution to the kidney developmental biology field. To my knowledge, there is no comparable tool existing in the literature and current transcriptome-wide spatial sequencing technologies cannot achieve the degree of spatial resolution offered by this integrated KSTAT approach. Further, I have not seen the authors' approach of combining scRNA-seq with targeted spatial sequencing reported in development of other organs beyond the kidney. Therefore, I predict this paper will be of high interest and utility to not only the kidney development community, but also extending to the broader readership of Development.

SUGGESTIONS TO AUTHORS

There are no major holes or flaws with the paper in its current form, and all the authors' conclusions are sufficiently supported by their data. The manuscript reads as a bit technical/computational for a Development paper, especially the Results section, but I suppose this is to be expected given the nature of the work. As a non-technical expert, I am still able to follow the logic and understand the conclusions even if not truly comprehending the methods. I have only a few minor suggestions that might improve the manuscript.

Minor recommendations

1. Most of the examples and analyses shown in the paper focus on the developing renal interstitium, which is a focus of the authors' lab. It might be useful to add some anecdotal data that also shows the projection of epithelial cell types or genes, which would likely be of interest to many readers and potential users of this Resource. For example, showing the localization of several nephron segments based on their clustering in the scRNA-seq data, analogous to what is shown in Fig. 4F-J. **The heterogeneity of the stroma is not well documented and, because our interest is in stromal heterogeneity, we focused on it. The heterogeneity of fibroblasts and mural cells we demonstrate here are completely novel observations and will be of great interest to the field. We absolutely agree with the reviewer that there are several other cell types with known but largely unexplored heterogeneity. Our dataset and computational tools will be of great value to investigators in these other fields and we hope they will take advantage of our resource to carry out future work. But we do not feel it is necessary for this publication and would only detract from future, focused efforts.**
2. Also, the UMAPs of the three scRNA-seq datasets with cluster annotations should be included in the supplement so that readers can understand all the cell types they should expect to be able to interrogate using KSTAT. **This has been included in supplemental figure 8 .**

3. Figure 5. Improved annotation of the figure to specify which images are E18.5 vs P3 would be helpful. This information is also not present in the figure legend, which is minimal. **This has been done.**

4. Labeling the panels (6C = LEF IF staining; 6F = EdU incorporation) in Figure 6 would enhance the reader's ability to interpret the figure. In Figure 7, directly labelling on the figure that white = bound receptors and yellow is unbound would be nice. **This has been done.**

5. Subheadings in the Results section would be helpful. **This has been done.**

Reviewer 3: SUMMARY OF THE ADVANCE MADE IN THIS PAPER AND ITS POTENTIAL SIGNIFICANCE TO THE FIELD

The manuscript by Cheney et al integrates single-nucleus RNA sequencing with high-resolution spatial transcriptomics with the aim of generating a comprehensive spatial and transcriptional atlas of embryonic and postnatal mouse kidneys. The authors introduce the Kidney Spatial Transcriptome Analysis Tool (KSTAT), which enables spatial mapping of gene expression, prediction of cell-cell communication, and pathway activity analysis, among other features. With this tool, the authors identify previously unappreciated heterogeneity among embryonic kidney pericytes. The authors generated this resource due to limitations of traditional transcriptomic approaches that lack spatial resolution and/or significant depth. Overall, this appears to be an effective approach to mapping single cell datasets back onto the kidney to generate predicted expression patterns and gain biological insights. While this approach could be useful for the broader developmental biology community, taking advantage of the numerous publicly available single cell/nucleus datasets, the resource is somewhat narrowly focused on the kidney and the author's interests. If the authors are unable to expand the analysis to include non-Interstitial biased in situ targets or other tissues, then it would still be useful to "market" this as a potentially more broadly applicable approach in the writing. Other notable concerns are included below.

SUGGESTIONS TO AUTHORS

1) The paper states that the authors developed a tool (Kidney Spatial Transcriptome Analysis Tool) and a virtual atlas. The impression given is that this might be a web resource that any user could utilize to look up their favorite gene, map a pathway, etc. and receive an output/image. What is the actual tool/resource-is it the method? Using the term "tool" feels like the reader should be provided something-some resource that is the result of the research performed in this manuscript. If there is no actual tool that can be identified, such as an online tool (like the Kidney Interactive Transcriptomics website) or code or GitHub that encompasses the entire pipeline (like a plug n play resource), the reviewer would recommend not naming this as a "tool" or resource but as a methodology they developed that could be applied to the kidney or other tissues. In that case, a step-by-step protocol that describes how each reader can make their own "resource" like the KSTAT would be of more significant utility.

Edit: This was written before using the "find" function on the manuscript and finding the GitHub link a last effort, buried in the Resource table near the end. This appears to be the resource generated but is not clear at all from reading the manuscript. **As mentioned above, we have a working website hosted locally. The original plan was to link to this resource through the GUDMAP resource but NIH has since de-funded GUDMAP. The costs of hosting such a website (over \$200k/year) are significant and we are attempting to find a sponsor. However, until we do, we cannot provide open access to our website. Instead, we can offer limited access to individuals on a collaborative basis. However, even without a publicly available website, the spatial data along with the accompanying single nucleus data sets also constitute a resource. The computational tool is also novel. The information provided on GitHub are sufficient for experienced users to regenerate our atlas. However, we have now included a step by step tutorial that allows less experienced researchers to reproduce the atlas in their own labs in GitHub.**

2) The Results seem somewhat of a rapid fire of what the resource can do, with short 2-3 sentence descriptions of what was performed. There is no significant depth to these with generally 1 example being presented each (ie G2 to M analysis, the Lef regulon, the metabolism of lipids

pathway). Each of these examples then nicely matches what is already known. While validation is important, showing one example is not very robust. Showing that the resource could provide novel biological insights, ie mapping a pathway or similar that is not well described in the kidney and then validating its significance through genetic or kidney culture experiments would significantly enhance the utility of this resource. This is a novel resource (the spatial atlas) and a novel computational tool. Access to these resources will facilitate hypothesis generation that can subsequently be tested using genetics and/or chemical approaches. Within the confines of a single manuscript, it is not possible to cover the spatial data along with the computational tool while also generating a hypothesis and testing that hypothesis. We consider this study to be the description of the resource and thus focused the manuscript on validating the resource. The reviewer will appreciate that we already exceeded the limits for a Development manuscript prior to addressing reviewer comments even with our lack of “significant depth”. Hopefully, readers will recognize the potential of our tool to generate and test novel hypotheses in the future. We have several projects currently being worked on in the lab that do use this tool to generate hypotheses that we are rigorously testing and we hope to publish on them soon. But we must first publish this tool.

3) The authors state that their methodology/dataset “offers unprecedented resolution and depth.” If this is the case, the authors should provide a direct comparison to other spatial transcriptomic analyses-how much more resolution and depth does it provide? Numbers would help support their claim. How does their methodology compare to a spatial technology such as Stereo-seq from Complete Genomics which can resolve single cells? Since submitting this manuscript, another study using the Cartana technology has been published and thus our data/approach is no longer “unprecedented”. Thus we have removed this term from the text.

4) The authors state that “This approach enabled us to map transcriptional data from each cell or nucleus onto a spatial framework, effectively generating a virtual in situ hybridization image for every gene expressed in the datasets.” To this reviewer it is unclear how every cell can be mapped, or what the coverage is. Perhaps in a single nucleus/cell sequencing dataset there are 3,000 interstitial cells that are sequenced. How many interstitial cells are then on an E18.5 section? Less than 3,000 or more than 3,000? If less, then how is every cell mapped and if more then are there cells on the section that don’t have a correlating single nucleus/cell? Maybe there is some misunderstanding on the reviewer’s behalf, but I think this should be clarified further in the text. We apologize for our lack of clarity. The mapping performed by our tool is probabilistic, not discrete. In other words we are mapping transcriptional clusters onto a tissue map based on the overlap between a single cell’s transcriptome and signals generated from our in situ sequencing of hallmark genes. We have added text to clarify this point in the methods and discussion of the revised manuscript. In addition, we have created a new version of figure s6 that we believe describes the technique more clearly.

5) Fig 3A should include higher mag views of the in situ and predicted patterns so the reader can more accurately see and assess the comparison. The current view is too low resolution even upon zooming to assess in what cells/structures these are expressed or predicted to be expressed. The quality of the images viewed by the reviewers is We can provide higher resolution figures but the reviewer must appreciate that, for technical reasons such diffusion of the stain, the resolution of standard colorimetric in situs is very low. It is nearly impossible to determine cell type specificity using standard section in situ hybridization without a priori knowledge. For instance, the medullary collecting ducts, thin descending loop of Henle, thin ascending loop of Henle and vasa recta all lie in the medullary region of the kidney and show epithelial/endothelial histology. One would have a very difficult time determining which of these cell types expressed a specific gene of interest based on in situ hybridization without some prior knowledge. Our tool combines the power of single cell transcriptomics with spatial data to overcome such limitations. Thus we do not feel that including this additional data is of significant value.

6) Results say E18.5 and P3 single nucleus data was generated but the Methods only describe E18.5. Methods have been update to include P3. It says that kidneys were snap frozen for genotyping. What genotyping needed to be performed on these samples? This was an error. Tissues were not genotyped. This has been corrected. There is no information provided on the strain of mice that were used. This needs to be added. Mice are on a mixed strain. This has been updated in the methods. Also, if the data generated is on a different background strain of mice than the publicly available datasets, does this matter at all for the tool or interpretation? Does this factor need to be

considered? We do not know the answer to this question as it would require performing the same analysis we have performed but using multiple different strains for both the single cell data and the in situ sequencing data. We have not done this and, to our knowledge, neither has anyone else. We are unaware of instances where different strains of mice show significantly different gene expression patterns from one embryo to another although specific examples of gene loss or gain exist. Such an example would result in a false positive or negative. For example, if one used single cell data from Bl6 mice and performed the spatial analysis on 129 mice and C57 mice did not express Tlr4, then when one queried Tlr4 expression, it would come up as not expressed in the C57 tissue because no Tlr4 expressing cell ever got mapped onto the spatial reference. In contrast, if 129 mice don't express Tlr4, it would still be mapped onto the spatial map based on co-expression of other genes. But it is important to note that the probability mapping is based on overlap in multiple hallmark genes (in our case, we had 84 successful mappings), which reduces the probability that mis-mapping of any one gene product will significantly affect performance. But this may be something that an investigator should consider when analyzing data, especially if they are looking at genes, pathways or processes that show differences between different strains. It should be noted that this same concern could/should be raised with any experiment that is not directly measuring expression in the tissue being analyzed (e.g. any analysis of publicly available RNA-Seq, scSeq, SnucSeq, Proteomics, Metabolomics datasets that were not performed in an isogenic or congenic background). We have added a some verbiage in the discussion covering this point and stating that the primary utility of this resource is to generate hypotheses and that all results should be independently validated.

7) Text states that Gucy1b1 was the only beta subunit detected at E18.5 and P3. But later it is stated that Gucylb2 overlaps with Gucyla2. Were both detected by the spatial profiling but not single cell data? This is not clear. The reference to Gucy1b2 was a typo and it has been corrected.

8) What is regulon activity for predicting transcription factor activity? This should be more clearly explained how this is done for readers who are not familiar. We have updated the text to describe this algorithm and provided a reference.

9) There is reference to the "statistical model used to predict likelihood of a cell's location." Is this a previously published method or did the authors generate this? It should be described better how this was developed. The derivation of the model is discussed in the methods and is based on previously published work from the Satija lab (cited in the methods section). We refer to this in the text of the updated version.

10) In Figure 1, why are Cited1 and Six2, which predominantly mark mesenchymal nephron progenitors classified as epithelial? This classification should be clarified so as to include epithelial progenitors or similar. Rather than provide only cluster identification, which has no inherent significance to the reader, we tried to break the figure up into broad categories to provide some biological insight. We struggled with how to name these categories. This categorization is imperfect. For instance, as the reviewer states, we included nephron progenitors in the epithelial category although these cells are not technically epithelial (although we also had long discussions about the definition of epithelia, which is actually surprisingly vague). If we included "nephron progenitor", then would we have to include "collecting duct progenitor", although these cells are most likely epithelial? In the end, we chose the categories we included because we felt they were sufficiently broad but also informative. We have clarified this in the revised text.

11) A schematic of the approach/technique should be added to help readers/researchers better understand what was necessary to generate the resource/tool-what does the pipeline look like? It took several readings to begin to understand all the experiments and analyses performed. Figure S9 was meant to be a schematic of the approach/technique and the pipeline but, based on reviewer feedback, it clearly did not achieve the intended goal. As mentioned above, we have provided a new schematic (figure s6) that we hope will help readers understand the broad concept behind the tool. A detailed figure describing the pipeline would be extensive and probably not of use to the normal Development reader. As mentioned above, we have now provided a detailed tutorial for "power users".

12) More specifics on when genes/things don't map properly should be provided. For example, for the 84 of 88 genes that had accurate spatial predictions, how off or wrong were the other genes?

We have provided explanations for why we believe the 4 “failed” hallmark genes did not accurately map. These are ofcourse specific examples and not meant to give a general impression of failure causes and rates. There is already a an entire paragraph in the discussion dedicated to potential reasons why a gene would be mapped inappropriately. There may be more reasons that we have not thought of. Hopefully the reviewer agrees that this is sufficient for this manuscript. In the future as more data is collected, it may be appropriate to extend this discussion in a review article.

13) Add subsections to the Results to help readers follow and break up sections. **This has been done.**

14) Figure legends for the main figures are very minimal and should be expanded. For example, Figure 5 legend is only a title. **This has been done.**

Second decision letter

MS ID#: dev.205003R1

MS TITLE: Integration of spatial and single nucleus transcriptomics to map gene expression in the developing mouse kidney

AUTHORS: Christopher Chaney, Alexandria N. Fusco, Elyse D. Grilli, Jane N. Warshaw, Peter M. Luo, Ondine Cleaver, Denise K. Marciano and Thomas J. Carroll

ARTICLE TYPE: Research Article

Dear Tom,

Many apologies for the delay. I am happy to tell you that your manuscript has been accepted for publication in Development, pending our standard publication integrity checks.

Reviewer 1

SUMMARY OF THE ADVANCE MADE IN THIS PAPER AND ITS POTENTIAL SIGNIFICANCE TO THE FIELD

The authors have developed a novel tool for spatial representation of transcriptomic data in the kidney.

SUGGESTIONS TO AUTHORS

All my questions have been answered.

Reviewer 2

SUMMARY OF THE ADVANCE MADE IN THIS PAPER AND ITS POTENTIAL SIGNIFICANCE TO THE FIELD

Chaney et al have revised their Resource manuscript to provide more clarity on some of the technical aspects related to development of the KSTAT tool, improve figure annotations and legends, and added some requested supplementary data. They also now provide a link to a step-by-step tutorial to expand accessibility of this Resource to the field. Again, I expect that the new KSTAT tool presented here will be a valuable and enabling resource for the kidney development community.

SUGGESTIONS TO AUTHORS

The authors have satisfactorily addressed all of my concerns.

Reviewer 3

SUMMARY OF THE ADVANCE MADE IN THIS PAPER AND ITS POTENTIAL SIGNIFICANCE TO THE FIELD

Provided in initial review

SUGGESTIONS TO AUTHORS

The authors have largely addressed reviewer concerns with changes/additions to the text which enhance the clarity of the manuscript. Additional details on the methodology, where to find resources, and the GitHub now make this more accessible to/usable by a broad range of researchers. The inability to provide zoomed images (utilizing the images that were already captured) for Fig 3A is still unclear, as some were provided for Figure 7 and would supply readers with enhanced visualization of signal comparison between the "real" in situ and the predicted/mapped pattern, even if discrete cell types cannot be discerned. But at minimum the authors should ensure that high resolution files are provided so that readers are able to zoom into the figures for a closer look without significant blurring. Also, it is unclear the source of the colorimetric in situ images. Did the authors generate them? This was not included in the methodology (see comment below). If they were obtained from another source like Gudmap this needs to be referenced appropriately.

Additional comments:

- 1) "Snap-frozen for genotyping" is still included in the methodology for the nuclei isolation and sequencing methodology. The response stated no genotyping was performed and this was removed.
- 2) There is a discrepancy between EdU labeling times between the text and Figure legend. Text states a 2-hour pulse but legend states a 1.5 hr pulse.
- 3) The methodology for EdU injection and detection, Lef1 immunofluorescence, colorimetric in situ hybridization, and multiplex fluorescent in situ hybridization are not described or referenced in the methods section. Sources of antibodies and in situ probes (if purchased) should also be noted.